# Step-Aware Residual-Guided Diffusion for EEG Spatial Super-Resolution

**Hongjun Liu[1]*  Leyu Zhou[1]*  Zijianghao Yang[1]    Chao Yao[2]†**
[1]School of Intelligence Science and Technology, University of Science and Technology Beijing
[2]School of Computer and Communication Engineering, University of Science and Technology Beijing

## Abstract

For real-world brain–computer interface (BCI) applications, lightweight Electroencephalography (EEG) systems offer the best cost–deployment balance. However, such spatial sparsity of EEG limits spatial fidelity, hurting learning and introducing bias. EEG spatial super-resolution methods aim to recover high-density EEG signals from sparse measurements, yet is often hindered by distribution shift and signal distortion and thus reducing fidelity and usability for EEG analysis and visualization. To overcome these challenges, we introduce SRGDiff, a step-aware residual-guided diffusion model that formulates EEG spatial super-resolution as dynamic conditional generation. Our key idea is to learn a dynamic residual condition from the low-density input that predicts the step-wise temporal and spatial details to add and uses the evolving cue to steer the denoising process toward high density reconstructions. At each denoising step, the proposed residual condition is additively fused with the previous denoiser feature maps, then a step-dependent affine modulation scales and shifts the activation to produce the current features. This iterative procedure dynamically extracts step-wise temporal rhythms and spatial-topographic cues to steer high-density recovery and maintain a fidelity–consistency balance. We adopt a comprehensive evaluation protocol spanning signal-, feature-, and downstream-level metrics across SEED, SEED-IV, and Localize-MI and multiple upsampling scales. SRGDiff consistently achieves higher SNR than the baseline ESTformer and STAD among Localize-MI, SEED and SEED-IV datasets, with up to roughly $75\%$ relative SNR improvement in the most challenging $8\times$ setting. Moreover, topographic visualizations comparison and substantial EEG-FID gains jointly indicate that our SR EEG mitigates the spatial–spectral shift between low- and high-density recordings. Our code is available at https://github.com/DhrLhj/ICLR2026SRGDiff.

## 1 Introduction

Electroencephalography (EEG) is a noninvasive technique for monitoring the brain's electrical activity, with widespread applications in neuroscience and clinical practice—ranging from brain–computer interfaces and epilepsy diagnosis to emotion recognition (Jiang et al., 2025). However, EEG's spatial resolution is inherently constrained by the number of scalp electrodes and the volume-conduction effect (Li et al., 2025a). High-density (HD) systems with hundreds of channels can mitigate these issues but are costly, cumbersome to deploy, and uncomfortable for extended wear, whereas low-density (LD) setups (e.g., 8 or 16 electrodes) are far more practical yet suffer from severe under-sampling bias (Wang et al., 2025). Indeed, as illustrated in Figure 1(c), the inter-channel activation patterns of 256-channel HD EEG diverge dramatically from those of 16-channel LD EEG, highlighting the strong bias in sparse recordings. EEG spatial super-resolution (SR) has therefore garnered growing attention, with methods that reconstruct high-density EEG from sparse recordings increasingly explored and applied.

Traditionally, EEG spatial super-resolution has relied on direct feature-mapping techniques that learn an end-to-end mapping from low-density to high-density representations. These methods fall

---

*Equal contribution.
†Corresponding author.

into two main categories: one employs convolutional neural networks or Transformers to upsample LD feature maps into HD ones (Tang et al., 2022), and the other leverages generative adversarial networks-based architectures that synthesize SR EEG signals conditioned on LD inputs (Wang et al., 2024). However, by treating the mapping as a static projection, these approaches often oversimplify the complex, nonlinear inter-electrode dependencies or demand vast amounts of training data and compute, resulting in overly smooth, detail-poor reconstructions that fail to capture true spatial consistency. Figure 1(c) indicates that such feature-mapping methods merely extend LD information, rather than recovering authentic HD channel relationships.

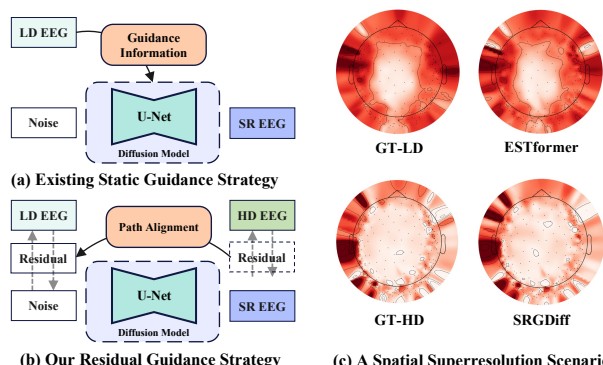

Figure 1: (a) Existing static guidance strategy vs. (b) our residual guidance strategy for EEG super-resolution, and (c) corresponding topographical maps of LD input, ESTformer output, GT HD EEG, and SRGDiff reconstruction.

Recently, diffusion models have been widely applied to time-series generation and missing-data imputation (Huang et al., 2025; Yuan & Qiao, 2024; Li et al., 2025b). In this context, EEG spatial super-resolution can be cast as conditional generation, where LD observations guide the recovery of HD signals. Within this line of work, researchers mainly focused on conditioning strategies through concatenating low-density features with the noise input (Vetter et al., 2024) or using cross-attention between modalities (Wang et al., 2025) as shown in Figure 1(a). While effective in practice, these approaches remain susceptible to a consistency–fidelity trade-off. Interpolation-oriented SR tends to cause distribution shift, making reconstructions adhere too closely to the LD observation and deviate from the HD ground truth. Conversely, generation-oriented SR often introduces distortion, producing HD-like content that fails to remain consistent with the LD input.

To tackle these challenges, we introduce Step-aware Residual-Guided Diffusion (SRGDiff) for EEG Spatial Super-Resolution, which reframes super-resolution as a dynamic conditional generation task. The core idea is to estimate the forward-noising residual from low-density channels, and use it as a per-step corrective direction in the reverse process. Technically, SRGDiff first encodes the low-density EEG with a pre-trained VAE encoder to obtain a compact latent and multi-scale features, and applies forward diffusion to the high-density latent. At each reverse step, a lightweight residual head predicts a path residual from the low-density features and uses it as a directional correction that is additively fused with the previous denoising features to form an incremental feature. The feature is then weighted with a step-dependent affine modulation estimated from the low-density features and the timestep embedding, yielding the current denoised features. This loop repeats over timesteps, coupling the low-density forward-noising and high-density reverse-denoising trajectories and progressively steering samples toward the high-density manifold. Our main contributions in this work can be summarized as follows:

- We recast EEG spatial super-resolution as **dynamic conditional generation**, coupling the LD forward–noising trajectory with the HD reverse–denoising trajectory to balance consistency with the LD observation and fidelity to the HD target.

- We propose a **dynamic residual guidance** paradigm: the path residual estimated from LD inputs serves as a per-step directional correction and is fused additively for incremental updates, yielding a stable, step-aware sampling scheme that remains effective across datasets and a wide range of SR factors.

- We establish a **three-level evaluation protocol** across three datasets, covering signal-level (temporal consistency, spectral fidelity, spatial topology), feature-level (representation quality), and downstream-level (classification accuracy), which provides a comprehensive assessment beyond pointwise error.

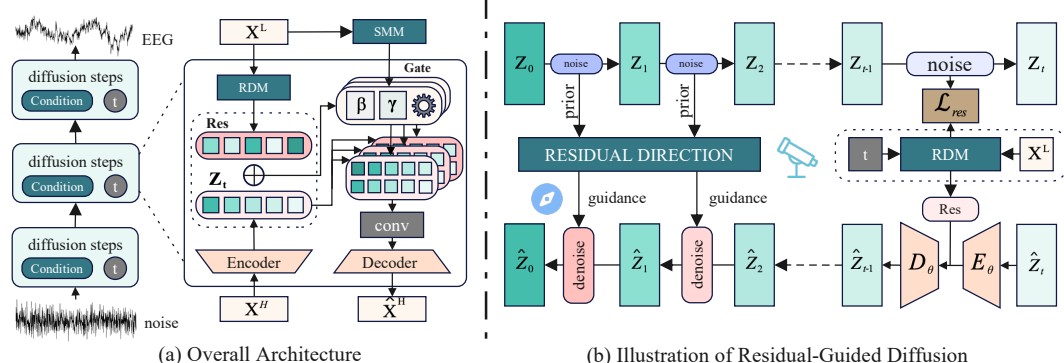

(a) Overall Architecture      (b) Illustration of Residual-Guided Diffusion

Figure 2: **SRGDiff overview.** (a) **Overall architecture:** Low-density EEG $X^L$ conditions the latent reverse process. RDM predicts a residual direction from $X^L$ and current decoder features. SMM provides step-aware affine parameters to fuse the residual and modulate activations. (b) **Residual learning:** At each step, the predicted residual guides denoising, and the residual derived from the forward noising process provides supervision via a residual loss.

## 2    RELATED WORK

**Diffusion Models for Missing Data Imputation.** Diffusion-based models have emerged as a powerful framework for time-series imputation, leveraging denoising diffusion processes to reconstruct missing values. Diffusion-TS (Yuan & Qiao, 2024) demonstrates interpretable conditional generation across diverse time-series without domain-specific priors. More recently, RDPI (Liu et al., 2025) further enhances precision and efficiency by first generating coarse estimates of missing values through deterministic interpolation, then conditioning a diffusion model on both observed data and these estimates to iteratively refine residual errors. SaSDim (Zhang et al., 2024) introduces self-adaptive noise scaling to preserve spatial dependencies within sensor networks, while SADI (Islam et al., 2025) integrate self-attention mechanisms to handle partial data missing.

**EEG Spatial Super-Resolution.** Early attempts at EEG spatial super-resolution adapted image-based frameworks to reconstruct dense electrode maps from sparse recordings. EEGSR-GAN (Corley & Huang, 2018) first applied adversarial training to hallucinate missing channels. EST-former (Li et al., 2025a) then introduced spatiotemporal transformers to model long-range dependencies across electrodes and successfully capture global patterns. More recent diffusion-based and attention-driven approaches have sought to address these limitations. DDPM-EEG (Vetter et al., 2024) leverages denoising diffusion probabilistic models to iteratively refine spatial patterns. STAD (Wang et al., 2025) tackles this by decomposing spatial–temporal interactions into spatial-temporal attention streams. This diffusion-based generative paradigm improves diversity and spectral fidelity compared to GANs, yet their static condition can still lead to distribution drifts and distortion.

**Residual Diffusion in Related Domains.** Several recent works incorporate residual signals into diffusion models. Ou et al. (2024) synthesize PET from MRI by learning a modality residual under prior information, i.e., a static cross-modal gap that conditions generation. Zhu et al. (2024) reconstruct event-driven video by predicting temporal residuals with inter-frame differences as the generation target to recover dynamics. Mao et al. (2025) address medical segmentation by learning a residual-to-prior that corrects a coarse segmentation, improving calibration and efficiency. These designs either treat the residual as a fixed target/offset or inject it once as a global prior, with weak coupling to the step-by-step reverse dynamics. In contrast, our method targets EEG spatial super-resolution and introduces a dynamic, step-aware residual direction that is re-estimated at every reverse step from the LD observation, and the timestep embedding.

## 3 PRELIMINARIES AND DYNAMIC CONDITIONAL FORMULATION

**Data and Latent Space.** Let $X^L \in \mathbb{R}^{C_L \times Length}$ and $X^H \in \mathbb{R}^{C_H \times Length}$ denote low- and high-density EEG with $C_H > C_L$. A pre-trained VAE encoder $E$ maps signals to a latent space $z = E(X^H)$ and a feature extractor $F$ provides LD features as condition $c = F(X^L)$ to condition generation. In practice, we reuse the VAE encoder as the feature extractor to obtain condition $c$.

**Forward Diffusion on the HD Latent.** We corrupt the HD latent with a standard Gaussian forward process

$$q(z_t \mid z_{t-1}) = \mathcal{N}\Big(\sqrt{\alpha_t}\, z_{t-1},\, (1 - \alpha_t)I\Big). \qquad t = 1, \ldots, T \tag{1}$$

**Dynamic Conditional Reverse Process.** In the reverse generation stage, sampling begins from an isotropic Gaussian noise initialization $\hat{z}_T \sim \mathcal{N}(0, I)$, and the model iteratively predicts $\hat{z}_{t-1}$ from $\hat{z}_t$ until recovering the final latent representation $\hat{z}_0$. Unlike conventional diffusion models that rely solely on a base denoiser, SRGDiff explicitly conditions each reverse step on low-density EEG observations, thereby coupling the LD forward-noising trajectory with the HD reverse-denoising trajectory. The reverse denoiser is defined as:

$$p_\theta(\hat{z}_{t-1} \mid \hat{z}_t, c) = \mathcal{N}\Big( \underbrace{\mu_\theta(\hat{z}_t, c)}_{\text{base denoiser}} + \underbrace{(\gamma_t, \beta_t, r_\phi(c, t))}_{\text{dynamic conditional update}} ,\, \beta_t I\Big). \tag{2}$$

Here, the base denoiser $\mu_\theta(\hat{z}_t, c)$ is implemented as a U-Net that estimates the noise component. To further enhance temporal fidelity and spatial coherence, we augment the base denoiser with two lightweight modules that inject step-wise conditional guidance from LD features:

- *Residual Direction Module (RDM).* At each timestep, RDM predicts a path residual $r_\phi(c, t)$ from the LD features and applies it as a directional correction:

$$\hat{z}_{t-1}^{RDM} = \hat{z}_t + r_\phi(c, t). \tag{3}$$

- *Step-Aware Modulation Module (SMM).* SMM calibrates the residual update with timestep-aware affine modulation. Specifically, it predicts a scale $\gamma_t$ and bias $\beta_t$ from LD features and the timestep embedding, and applies them to the residual-corrected state:

$$\hat{z}_{t-1}^{SMM} = \gamma_t \odot \hat{z}_{t-1}^{RDM} + \beta_t. \tag{4}$$

Together, the pair $(r_\phi, \gamma_t, \beta_t)$ realizes dynamic conditional generation: at every denoising step, LD features provide both a directional residual and a step-dependent modulation strength, yielding a stable and temporally consistent correction of the reverse diffusion process.

## 4 PROPOSED METHOD

This section presents the step-aware residual-guided diffusion framework, and outlines its architecture and core components. SRGDiff reframes SR as dynamic conditional generation that couples the LD forward-noising trajectory with the HD reverse-denoising trajectory, using an LD-estimated residual as a per-step corrective direction and a step-aware calibration to modulate its strength. The framework comprises four parts: the latent diffusion model backbone, Residual Direction Module (RDM) for additive directional updates, Step-Aware Modulation Module (SMM) for step-dependent modulation, and the overall training strategy. An overview is provided in Figure 2, and the following subsections describe each component in detail.

### 4.1 LATENT DIFFUSION MODEL BACKBONE

Our backbone consists of a VAE that builds the latent space and a denoising U-Net that performs diffusion in that space. The VAE follows the EEG autoencoding setup of Aristimunha et al. (2023).

We train an encoder–decoder $(E, D)$ on HD EEG $X^H$ to obtain $z = E(X^H)$ and $\widehat{X}^H = D(\widehat{z})$, and optimize a reconstruction–regularization objective

$$\mathcal{L}_{\text{VAE}} = \|\widehat{X}^H - X^H\|_2^2 + \lambda_{\text{spec}}\|\text{STFT}(\widehat{X}^H) - \text{STFT}(X^H)\|_1 + \lambda_{KL} \text{KL}\big(q_E(z \mid X^H) \| \mathcal{N}(0, I)\big), \tag{5}$$

where $\text{STFT}(\cdot)$ denotes the short-time Fourier transform applied along the temporal dimension of each EEG channel to encourage spectral fidelity.

Empirically, we set the spectral weight to $0.1$ and the KL weight to $10^{-4}$. After convergence, $(E, D)$ are frozen. On top of this latent space, we adopt a latent-diffusion model in the style of Rombach et al. (2022).

## 4.2 RESIDUAL DIRECTION MODULE

In the context of EEG spatial SR, most diffusion approaches condition the U-Net via feature concatenation or cross-attention. We instead turn EEG spatial SR into finer and step-aware conditioning by learning residual direction from low-density recordings. We use the VAE encoder to extract multi-scale condition $c$ and an RDM head $R_\phi$ takes $(c, \tau(t))$ to predict a residual $Res_t$ in the encoder feature space, which acts as a per-step directional correction to the reverse process. Concretely, we first sample a timestep $t$, encode the HD EEG to obtain the latent $z_0 = E(X^H)$, and draw

$$z_t = \sqrt{\bar{\alpha}_t}\, z_0 + \sqrt{1 - \bar{\alpha}_t}\, \epsilon, \qquad \epsilon \sim \mathcal{N}(0, I). \tag{6}$$

In the forward process, we obtain at each step the noise-corrupted latent of the HD EEG and use this sequence of step-dependent features as supervision targets for the residual. The residual labels span $t = 0, \ldots, T$ and are defined as $\delta z_t := z_0 - z_t$.

In the reverse process, we estimate $\delta z_t$ from the low-density EEG to supply the step-wise temporal and spatial details required for denoising. We introduce a lightweight convolutional predictor $R_\phi$ that takes the timestep embedding $\tau(t)$ and LD features $c = F(X^L)$ as input and outputs the residual feature $Res_t$, and trained by

$$Res_t = R_\phi\big(\tau(t), c\big), \quad \mathcal{L}_{\text{res}} = \sum_{t=0}^{T} \big\| Res_t - \delta z_t \big\|_2^2. \tag{7}$$

Finally, the predicted residual feature is then added to $\hat{z}_t$ as an incremental update:

$$\hat{z}_t^{RDM} = LayerNorm(\hat{z}_t) + Res_t. \tag{8}$$

## 4.3 STEP-AWARE MODULATION MODULE

After obtaining $\hat{z}_t^{\text{RDM}}$, we further modulate the current step to control the extent to which the residual condition influences denoising. To enforce temporal fidelity, SMM explicitly weights the current diffusion timestep with a step-dependent affine modulation estimated from the low-density features and the current timestep embedding.

Specifically, SMM first encodes the low-density EEG through a lightweight 1D convolutional network $E_{SMM}$ to produce a feature map $h_t$. To enable the conditioning to recognize the current diffusion step, SMM maps each sampled timestep $t$ into a sinusoidal time embedding $e_t$. A learnable weight $\sigma_t$ that decays linearly with $t$ balances these two streams, yielding a fused feature:

$$\widetilde{h}_t = \sigma_t h_t + (1 - \sigma_t)e_t = \sigma_t E_{SMM}(c) + (1 - \sigma_t)e_t. \tag{9}$$

For spatial coherence, we adopt an affine calibration mechanism. The fused feature $\widetilde{h}_t$ is passed through two MLPs $MLP_\gamma$ and $MLP_\beta$ to predict channel-wise scale $\gamma_t^c$ and bias $\beta_t^c$:

$$\hat{z}_t^{SMM} = \gamma_t \odot \hat{z}_t^{RDM} + \beta_t^c = MLP_\gamma(\widetilde{h}_t, t) \odot \hat{z}_t^{RDM} + MLP_\beta(\widetilde{h}_t, t). \tag{10}$$

Finally, SRGDiff feeds the updated latent $\hat{z}_t$ into the U-Net decoder to obtain the next denoised state $\hat{z}_{t-1}$.

## 4.4 TRAINING STRATEGY

To stabilize optimization and decouple latent representation learning from conditional diffusion modeling, we adopt a two-stage training strategy.

**Stage 1: VAE Pre-training.** We first train the VAE encoder-decoder pair on high-density EEG data to obtain a stable and structured latent space. After convergence, the VAE parameters are frozen to provide fixed latent representations for subsequent diffusion modeling.

**Stage 2: Residual-Guided Latent Diffusion.** On the frozen latent space, the final training objective is a weighted combination of the three terms:

$$\mathcal{L}_{\text{Stage 2}} = \mathbb{E}_{z_0,\epsilon,t}\big[\|\epsilon-\epsilon_\theta(z_t,t,c)\|_2^2\big] + \lambda_{res}\sum_{t=1}^{T}\|R_\varphi(c,t)-(z_0-z_t)\|_2^2 + \lambda_{SMM}(\|\gamma_t-1\|_2^2+\|\beta_t\|_2^2).$$
(11)

Empirically, we set the residual weight to $1$ and the SMM weight to $10^{-2}$. The term $\lambda_{\text{SMM}}\big(\|\gamma_t - 1\|_2^2 + \|\beta_t\|_2^2\big)$ serves as a regularization component to prevent excessively large values of $\gamma_t$ and $\beta_t$, thereby stabilizing the training dynamics.

## 5 EXPERIMENTS

### 5.1 DOWNSTREAM DATASETS

In this study, we employ three publicly available EEG datasets. The **SEED dataset** (Zheng & Lu, 2015) uses 15 film clips of approximately four minutes each as emotional stimuli to induce stable and continuous emotional responses in three categories: positive, neutral, and negative. Data were acquired via 62 channels at 1000 Hz, downsampled to 200 Hz, band-pass filtered (0–75 Hz) and segments with faulty sensors removed. **SEED-IV** (Zheng et al., 2018) extends SEED by using the same 15 subjects and 62-channel setup (1000 Hz) but adds music and image stimuli to evoke happiness, sadness, fear and neutral states; preprocessing mirrors that of SEED. **Localize-MI** (Mikulan et al., 2020) contains 61 presurgical sessions from seven drug-resistant epilepsy patients, where 256-channel scalp EEG was recorded at 8000 Hz during 0.1–5 mA intracerebral single-pulse stimulation; preprocessing includes a 0.1 Hz high-pass filter, notch filter, bad-channel/trial removal and alignment of trials to the -300 ms to +50 ms stimulus-artifact window.

### 5.2 EXPERIMENT SETUP

**Data Preprocessing.** The experimental setup for the EEG super-resolution task follows the ES-Tformer and STAD frameworks. The preprocessed EEG signals were segmented into fixed-length windows: continuous, non-overlapping 4-second windows for SEED and SEED-IV datasets, and 260 ms windows (from 250 ms before stimulation to 10 ms after) for Localize-MI. In SEED and SEED-IV, we designed different super-resolution scale factors ($2\times$, $4\times$ and $8\times$) to evaluate reconstruction performance. The selection of visible channels and the super-resolution scaling factors follow the configurations used in ESTformer. For Localize-MI, due to the high channel density, we applied more extensive scale factors ($2\times$, $4\times$, $8\times$, $16\times$).

**Training & Environment Settings.** For each dataset, we split the data into train/test with an $80\%/20\%$ ratio and reserve $10\%$ of the training portion as a validation set. Stage I is trained only on the HD signals from the training split. Stage II is trained on paired LD/HD samples constructed from the same training split by masking HD channels according to the target SR scale; the validation set is used for early stopping and hyperparameter selection. The held-out test split is used once for final reporting, with no fine-tuning.

**Baselines.** We compare SRGDiff with strong EEG SR and time-series imputation baselines: **ES-Tformer** (Li et al., 2025a) and **STAD** (Wang et al., 2025) (transformer-/diffusion-based EEG SR), **DDPMEEG** (Vetter et al., 2024) (diffusion for ECoG SR), **SaSDim** (Zhang et al., 2024) and **SADI** (Islam et al., 2025) (advanced missing data imputation), and the two-stage residual method **RDPI** (Liu et al., 2025). We use authors' official implementations when available and otherwise provide

| Model | Ref | Metric | 2 | 4 | 8 | 16 |
|---|---|---|---|---|---|---|
| SaSDim | IJCAI 2024 | NMSE | 0.2675±0.003 | 0.3427±0.001 | 0.4174±0.004 | 0.4613±0.003 |
| | | PCC | 0.8194±0.002 | 0.7246±0.007 | 0.6926±0.003 | 0.6476±0.002 |
| | | SNR | 5.7443±0.007 | 4.3796±0.003 | 3.5549±0.009 | 2.7678±0.005 |
| SADI | AAAI 2025 | NMSE | 0.2637±0.003 | 0.3442±0.001 | 0.4164±0.004 | 0.4566±0.003 |
| | | PCC | 0.8243±0.002 | 0.7391±0.007 | 0.6944±0.003 | 0.6554±0.002 |
| | | SNR | 5.7511±0.007 | 4.3724±0.003 | 3.5498±0.008 | 2.8942±0.009 |
| RDPI | AAAI 2025 | NMSE | 0.2561±0.003 | 0.3562±0.001 | 0.4076±0.004 | 0.4531±0.003 |
| | | PCC | 0.8246±0.002 | 0.7396±0.007 | 0.7062±0.003 | 0.6549±0.002 |
| | | SNR | 5.7311±0.007 | 4.3966±0.003 | 3.5643±0.009 | 2.7731±0.007 |
| DDPMEEG | Patterns 2024 | NMSE | 0.2046±0.003 | 0.3108±0.001 | 0.3554±0.004 | 0.4076±0.002 |
| | | PCC | 0.8516±0.002 | 0.8163±0.007 | 0.7306±0.003 | 0.6739±0.002 |
| | | SNR | 6.2151±0.008 | 5.5126±0.003 | 3.9891±0.009 | 3.2715±0.005 |
| ESTformer | KBS 2025 | NMSE | 0.2721±0.003 | 0.3578±0.001 | 0.4466±0.004 | 0.4837±0.002 |
| | | PCC | 0.8061±0.002 | 0.7205±0.007 | 0.6867±0.003 | 0.6319±0.002 |
| | | SNR | 5.5403±0.008 | 3.8671±0.003 | 3.3023±0.007 | 2.5671±0.004 |
| STAD | TCE 2025 | NMSE | 0.1902±0.003 | 0.3067±0.001 | 0.3649±0.004 | 0.4106±0.003 |
| | | PCC | 0.8635±0.002 | 0.8194±0.007 | 0.7216±0.003 | 0.6694±0.002 |
| | | SNR | 7.2591±0.008 | 5.5234±0.003 | 3.8715±0.009 | 3.2642±0.005 |
| SRGDiff | OURS | NMSE | **0.1449**±0.003 | **0.2384**±0.001 | **0.2957**±0.004 | **0.3457**±0.002 |
| | | PCC | **0.9213**±0.002 | **0.8854**±0.007 | **0.8323**±0.003 | **0.7322**±0.002 |
| | | SNR | **8.3755**±0.008 | **6.3617**±0.003 | **5.2249**±0.009 | **4.0197**±0.006 |

Table 1: Performance of all methods on Localize-MI across different channel settings.

carefully verified reimplementations, applying their recommended hyperparameters and unifying training epochs and sampling steps across methods.

## 5.3 EVALUATION PROTOCOL.

We assess SR quality at three complementary levels to balance faithfulness to the ground truth, preservation of neurophysiological structure, and practical utility.

**Signal level** (does the waveform match?): We follow **ESTformer** and report normalized mean squared error (NMSE), Pearson correlation coefficient (PCC), and reconstruction signal-to-noise ratio (SNR) with respect to the HD reference, plus topology maps for qualitative inspection (formal definitions in the Appendix).

**Feature level** (does the representation distribution match?): We adopt **EEG-FID** following Lai et al. (2025), using a frozen EEGNet trained per dataset on its training split; the embedding dimension is 256 for SEED/SEED-IV and 512 for Localize-MI. In addition, we report a **frequency-domain MAE**: we first compute channel-wise STFTs of the reconstructed and reference HD EEG, form their power spectra, and then compute the normalized mean squared error between these spectra averaged over channels and frequency bins, so as to capture spectral distortions that are not reflected by time-domain NMSE alone.

**Downstream level** (is it useful?): We evaluate SEED/SEED-IV subject-dependent emotion recognition without cross-validation and binary epileptic classification on Localize-MI, both reporting accuracy. All results are summarized as mean±std over subjects; implementation details and metric formulas are provided in the Appendix.

## 5.4 MAIN RESULTS

We report signal-level reconstruction quality in Tables 1 and 2. For the most demanding $16\times$ up-sampling on Localize-MI, SRGDiff attains an NMSE of 0.3457, over 15% lower than DDPMEEG's 0.4076, indicating that dynamic conditioning effectively guides the diffusion model to generate

| Model | Metric | SEED (62) | | | SEED-IV (62) | | |
|---|---|---|---|---|---|---|---|
| | | 2 | 4 | 8 | 2 | 4 | 8 |
| SaSDim | NMSE | 0.4399±0.004 | 0.6234±0.002 | 0.7767±0.007 | 0.3633±0.004 | 0.5543±0.002 | 0.7122±0.005 |
| | PCC | 0.7341±0.002 | 0.5649±0.001 | 0.4349±0.004 | 0.7249±0.002 | 0.6211±0.009 | 0.5009±0.003 |
| | SNR | 4.1154±0.096 | 2.2940±0.046 | 1.1349±0.127 | 4.5940±0.009 | 2.6004±0.004 | 1.6211±0.111 |
| SADI | NMSE | 0.4439±0.004 | 0.6049±0.002 | 0.8106±0.007 | 0.3557±0.004 | 0.5349±0.002 | 0.6844±0.005 |
| | PCC | 0.7234±0.002 | 0.5819±0.001 | 0.4064±0.004 | 0.7624±0.002 | 0.6293±0.009 | 0.5243±0.003 |
| | SNR | 4.2419±0.097 | 2.5160±0.046 | 1.0137±0.127 | 4.7093±0.009 | 2.6044±0.004 | 1.6610±0.112 |
| RDPI | NMSE | 0.4064±0.004 | 0.6134±0.002 | 0.7916±0.007 | 0.3491±0.004 | 0.5416±0.002 | 0.6915±0.005 |
| | PCC | 0.7416±0.002 | 0.5716±0.001 | 0.4216±0.004 | 0.7861±0.002 | 0.6316±0.009 | 0.5164±0.003 |
| | SNR | 4.2619±0.097 | 2.3160±0.046 | 1.0316±0.127 | 4.7190±0.009 | 2.6194±0.004 | 1.6492±0.115 |
| DDPMEEG | NMSE | 0.4916±0.004 | 0.7319±0.002 | 0.8634±0.007 | 0.5136±0.004 | 0.6513±0.001 | 0.7916±0.005 |
| | PCC | 0.6941±0.002 | 0.5134±0.001 | 0.3419±0.004 | 0.7346±0.001 | 0.5316±0.009 | 0.4305±0.003 |
| | SNR | 4.1943±0.095 | 1.5391±0.042 | 0.9431±0.125 | 4.4165±0.008 | 2.1064±0.003 | 1.6105±0.110 |
| ESTformer | NMSE | 0.3288±0.004 | 0.3483±0.002 | 0.4149±0.007 | 0.3448±0.004 | 0.3911±0.0015 | 0.5125±0.005 |
| | PCC | 0.8368±0.002 | 0.8012±0.001 | 0.7670±0.004 | 0.8106±0.002 | 0.7822±0.009 | 0.7048±0.003 |
| | SNR | 5.0560±0.097 | 4.5838±0.044 | 3.8871±0.126 | 4.7535±0.008 | 4.1933±0.003 | 2.9821±0.113 |
| STAD | NMSE | 0.4319±0.004 | 0.6913±0.002 | 0.8671±0.007 | 0.3819±0.004 | 0.6713±0.002 | 0.7193±0.005 |
| | PCC | 0.7136±0.002 | 0.4946±0.001 | 0.3441±0.004 | 0.7316±0.002 | 0.5219±0.009 | 0.4319±0.003 |
| | SNR | 4.1364±0.099 | 1.4349±0.043 | 0.9134±0.125 | 4.4930±0.008 | 2.0492±0.003 | 1.6193±0.114 |
| SRGDiff | NMSE | **0.1632**±0.004 | **0.2977**±0.002 | **0.3494**±0.007 | **0.1663**±0.004 | **0.2115**±0.002 | **0.2603**±0.005 |
| | PCC | **0.9102**±0.002 | **0.8445**±0.001 | **0.8167**±0.004 | **0.9113**±0.002 | **0.8846**±0.009 | **0.8210**±0.003 |
| | SNR | **7.8413**±0.097 | **5.2606**±0.043 | **4.5912**±0.127 | **7.8660**±0.008 | **6.6402**±0.003 | **6.0346**±0.120 |

Table 2: Performance comparison of different models on SEED and SEED-IV datasets across different channel settings.

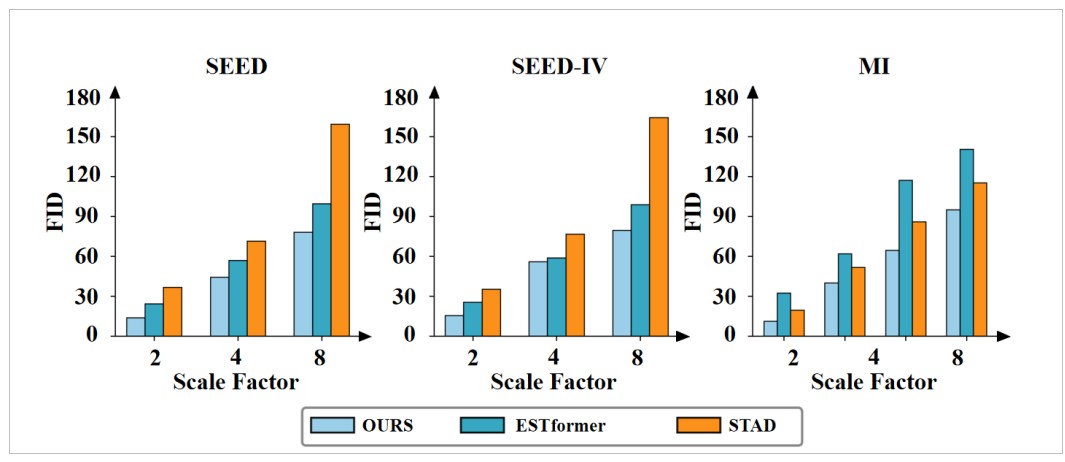

Figure 3: EEG-FID evaluation across three datasets compared with ESTformer and STAD.

super-resolved signals that closely approximate the true high-density data. Its PCC improves from 0.6739 to 0.7322, and its SNR increases from 3.27 dB to 4.02 dB (over 22%), demonstrating that under challenging settings the dynamic conditioning still learns the HD trend while maintaining a favorable signal-to-noise ratio. On SEED with high temporal variability and frequent outliers, SRGDiff reduces the $2\times$ NMSE from ESTformer's 0.3288 to 0.1632 (a reduction of more than 50%) and raises PCC from 0.8368 to 0.9102. A similar pattern is observed on SEED-IV, where NMSE drops to 0.1663 versus 0.3448 and PCC increases to 0.9113 versus 0.8106, indicating that despite lower SNR, the dynamic conditioning exhibits strong generalization.

## 5.5 FEATURE-LEVEL RECONSTRUCTION EVALUATION

We report EEG-FID results in Figure 3, and our method consistently achieves the lowest FID scores across SEED, SEED-IV, and Localize-MI datasets under different scale factors. These results indi-

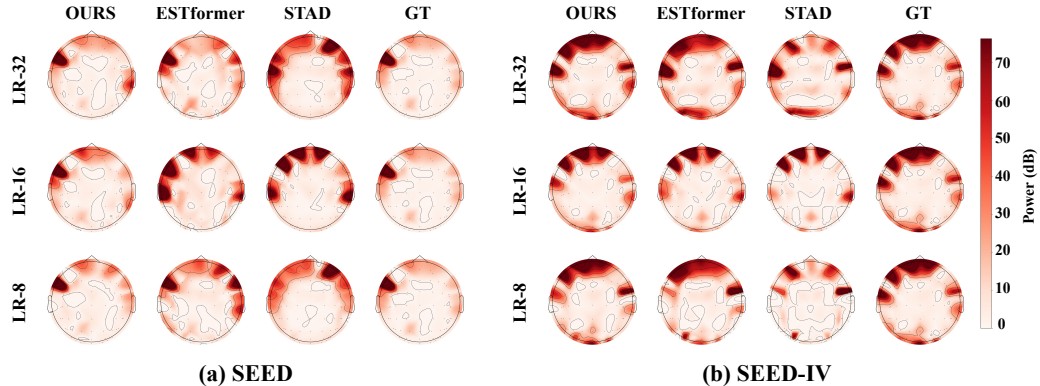

**(a) SEED**                     **(b) SEED-IV**

Figure 4: Visualization of EEG topographic maps between ground-truth and reconstructed EEG signals by ESTformer, STAD and SRGDiff.

| Model | SEED | | | SEED-IV | | | Localize-MI | | | |
|---|---|---|---|---|---|---|---|---|---|---|
| | 2× | 4× | 8× | 2× | 4× | 8× | 2× | 4× | 8× | 16× |
| ESTformer | 6.96 | 9.31 | 9.73 | 7.11 | 8.30 | 8.86 | 7.03 | 16.67 | 32.32 | 35.73 |
| STAD | 9.19 | 11.04 | 14.40 | 9.50 | 10.95 | 13.12 | 8.76 | 13.11 | 22.53 | 25.37 |
| **SRGDiff** | **3.89** | **5.12** | **4.95** | **3.99** | **4.08** | **4.84** | **3.86** | **7.30** | **11.50** | **13.76** |

Table 3: Frequency-domain MAE between reconstructed and real HD topomaps on SEED, SEED-IV, and Localize-MI under different SR factors.

cate that our approach generates EEG signals that are statistically closer to the real distribution in the temporal domain.

We further analyze the spectral fidelity of generated signals by visualizing EEG topographic maps under different scale factors. An EEG topographic map projects the power spectral density (PSD) of each channel onto the scalp surface, providing an intuitive representation of the spatial distribution of oscillatory energy. As shown in Figure 4 and Figure 15, although our reconstructed signals still exhibit minor deviations from the original data, they preserve a high degree of overlap in critical regions with strong PSD responses.

Beyond qualitative inspection, we also report a frequency-domain error metric that quantifies the mean absolute error between reconstructed and real HD topomaps. As shown in Table 3, SRGDiff consistently achieves the lowest frequency-domain MAE across datasets and SR scales, indicating better preservation of the spatial distribution of spectral power.

## 5.6 DOWNSTREAM TASKS

Table 4 reports results on the three datasets under various super-resolution scales. As the scale grows, accuracy for both raw and super-resolved inputs declines, yet SRGDiff's reconstructions consistently maintain a clear advantage. In particular, SRGDiff greatly outperforms missing-value imputation methods and leads all other spatial imputation approaches. At 2× scale factor, its classification accuracy approaches that obtained from the original full-channel recordings. We further compared the runtime efficiency of different methods as shown in Table 13. Although our proposed SRGDiff is slightly slower than the transformer-based ESTformer, it can still complete EEG super-resolution within 0.1s, which meets the real-time requirement in practical applications. The detailed runtime statistics of all models are provided in the Appendix.

## 5.7 ABLATION STUDY

To evaluate the contribution of each module in SRGDiff to EEG super-resolution reconstruction, we conducted ablation studies comparing SRGDiff with three variant models. **LDM+LD** keeps only

| Method | SEED | | | SEED-IV | | | Localize-MI | | | |
|---|---|---|---|---|---|---|---|---|---|---|
| | 2 | 4 | 8 | 2 | 4 | 8 | 2 | 4 | 8 | 16 |
| GT | 0.7152 | 0.7152 | 0.7152 | 0.7027 | 0.7027 | 0.7027 | 0.8368 | 0.8368 | 0.8368 | 0.8368 |
| LR | 0.4981 | 0.4702 | 0.4424 | 0.5685 | 0.5618 | 0.5011 | 0.7208 | 0.6534 | 0.5219 | 0.3862 |
| SaSDim | 0.5097 | 0.4793 | 0.4429 | 0.5794 | 0.5692 | 0.4912 | 0.7193 | 0.6519 | 0.5237 | 0.3845 |
| SADI | 0.5137 | 0.4834 | 0.4456 | 0.5718 | 0.5644 | 0.4987 | 0.7215 | 0.6634 | 0.5314 | 0.3957 |
| RDPI | 0.5044 | 0.4802 | 0.4531 | 0.5591 | 0.5741 | 0.5071 | 0.7230 | 0.6624 | 0.5210 | 0.3892 |
| DDPMEEG | 0.4738 | 0.4610 | 0.4238 | 0.5548 | 0.5487 | 0.4838 | 0.7015 | 0.6387 | 0.5187 | 0.3767 |
| ESTformer | 0.6887 | 0.6509 | 0.6057 | 0.6782 | 0.6500 | 0.5084 | 0.7445 | 0.6033 | 0.4739 | 0.4391 |
| STAD | 0.5437 | 0.5249 | 0.4610 | 0.6651 | 0.6410 | 0.4977 | 0.7589 | 0.6797 | 0.6344 | 0.5384 |
| SRGDiff | **0.7019** | **0.6812** | **0.6273** | **0.6821** | **0.6558** | **0.5127** | **0.7641** | **0.7163** | **0.6806** | **0.5887** |

Table 4: Classification accuracy comparison across different methods and datasets. GT represents the ground truth performance. Best results are shown in bold, second-best are underlined.

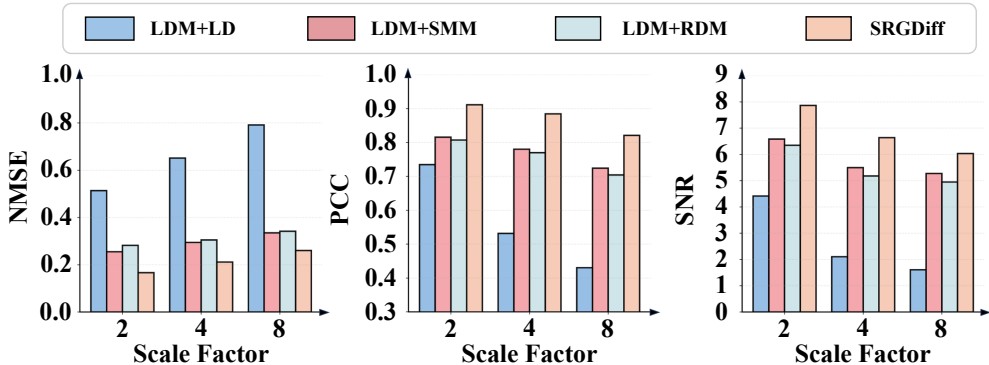

Figure 5: Ablation study performance comparison between SRGDiff and three variant models on the SEED dataset.

the VAE–DDIM backbone and takes the LD EEG as its input condition; **LDM+SMM** preserves the Step-aware modulation module; **LDM+RDM** retains the Residual Direction Module. All models were tested under the same experimental settings.

Figure 5 reports NMSE, PCC and SNR across $2\times$, $4\times$ and $8\times$ upsampling in SEED dataset. At $8\times$, adding SMM to the baseline cuts NMSE from $0.86$ to $0.45$ with $47\%$ reduction and boosts PCC from $0.34$ to $0.69$, demonstrating its effectiveness in temporally aligning the denoising trajectory. Incorporating RDM yields a comparable NMSE reduction with $44\%$ and raises PCC to $0.67$, highlighting its role in injecting prior information for spatial consistency. When combined in SRGDiff, these modules further decrease NMSE to $0.34$ with $60\%$ overall reduction and elevate PCC to $0.81$. More ablation results in SEED-IV and Localize-MI datasets are shown in Figure 11 in the Appendix.

## 6 CONCLUSION

We introduced SRGDiff, a step-aware residual-guided diffusion model that reframes EEG spatial super-resolution as guided HD generation with a step-aware residual direction and adaptive modulation. Across SEED, SEED-IV, and Localize-MI, SRGDiff consistently improves signal-level metrics, achieves the best EEG-FID across scales, and better preserves spectral and scalp topographies. Downstream evaluations further show higher accuracy on emotion recognition and patient-wise classification, indicating that the reconstructed signals are not only visually and statistically closer to HD EEG but also more useful for analysis. These results validate that explicit, step-wise conditioning on sparse inputs is both necessary and effective for high-fidelity, topology-preserving EEG super-resolution.

## 7 ACKNOWLEDGEMENTS

This work was supported by the National Key Research and Development Program of China (Grant No. 2022ZD0118001), the National Natural Science Foundation of China (Grant Nos. U22A2022, 62332017), and was conducted in part at the MOE Key Laboratory of Advanced Materials and Devices for Post-Moore Chips, the Beijing Key Laboratory of Big Data Innovation and Application for Skeletal Health Medical Care, and the Beijing Advanced Innovation Center for Materials Genome Engineering.

## 8 ETHICS STATEMENT

This work adheres to the ICLR Code of Ethics. In this study, no human subjects or animal experimentation was involved. All datasets used, including SEED, SEED-IV and Localize-MI, were sourced in compliance with relevant usage guidelines, ensuring no violation of privacy. Details could be found in Appendix. We have taken care to avoid any biases or discriminatory outcomes in our research process. No personally identifiable information was used, and no experiments were conducted that could raise privacy or security concerns. We are committed to maintaining transparency and integrity throughout the research process.

## 9 REPRODUCIBILITY STATEMENT

We have made every effort to ensure that the results presented in this paper are reproducible. All code and datasets have been made publicly available in an anonymous repository to facilitate replication and verification. The experimental setup, including training steps, model configurations, and hardware details, is described in detail in the paper. We have also provided a full description of SRGDiff, to assist others in reproducing our experiments.

We believe these measures will enable other researchers to reproduce our work and further advance the field.

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
