# APPENDIX FOR STEP-AWARE RESIDUAL-GUIDED DIFFUSION FOR EEG SPATIAL SUPER-RESOLUTION

## 1    LLM USAGE

Large Language Models (LLMs) were used to aid in the writing and polishing of the manuscript. Specifically, we used an LLM to assist in refining the language, improving readability, and ensuring clarity in various sections of the paper. The model helped with tasks such as sentence rephrasing, grammar checking, and enhancing the overall flow of the text.

It is important to note that the LLM was not involved in the ideation, research methodology, or experimental design. All research concepts, ideas, and analyses were developed and conducted by the authors. The contributions of the LLM were solely focused on improving the linguistic quality of the paper, with no involvement in the scientific content or data analysis.

The authors take full responsibility for the content of the manuscript, including any text generated or polished by the LLM. We have ensured that the LLM-generated text adheres to ethical guidelines and does not contribute to plagiarism or scientific misconduct.

## 2    DATASET PREPROCESS DETAILS

### 2.1    DATASET DETAILS

We use three publicly available EEG datasets: SEED, SEED-IV, and Localize-MI.

**SEED** (available at `http://bcmi.sjtu.edu.cn/~seed/`) consists of recordings from 15 subjects watching emotion-eliciting film clips ($\approx$ 4 min each) designed to induce positive, neutral, and negative states. Data were acquired with a 62-channel 10–20 montage at 1000 Hz, downsampled to 200 Hz, and band-pass filtered to 0.5–75 Hz. We manually inspected and removed sessions with sensor faults as depicted in Figure 1 and Figure 2.

**SEED-IV** (available at `http://bcmi.sjtu.edu.cn/~seed/seed-iv.html`) is a publicly available EEG dataset designed for emotion recognition research. It includes four emotional categories: happiness, sadness, neutrality, and fear. Emotional states are elicited using two types of stimuli: music and images. EEG recordings were collected from 15 subjects using a 62-channel system with a sampling rate of 1000 Hz.During preprocessing, the EEG signals were downsampled to 200 Hz and filtered with a bandpass filter ranging from 0.5 to 75 Hz. To ensure data quality, visually corrupted or invalid trials were manually excluded.

**Localize-MI** (available at `https://doi.org/10.12751/g-node.1cc1ae`) is a high-density intracranial EEG dataset from seven drug-resistant epilepsy patients during 61 presurgical sessions. Stereo–EEG electrodes delivered single-pulse biphasic currents (0.1–5 mA), and 256 channels were recorded at 8000 Hz. Preprocessing included 0.1 Hz high-pass filtering, notch filters at 50/100/150/200 Hz, bad-channel/trial removal, and trial alignment using stimulation artifact peaks (−300 to +50 ms window). In the Localize-MI dataset, we designed a binary classification task (epileptic vs. nonepileptic) to evaluate the effectiveness of synthetic super-resolution EEG (SR EEG) in detecting epileptic abnormalities. Specifically, EEG signals recorded before electrical stimulation are labeled as nonepileptic, while those recorded during stimulation are labeled as epileptic. The detailed experimental setup follows the description provided in the STAD **?** model section.

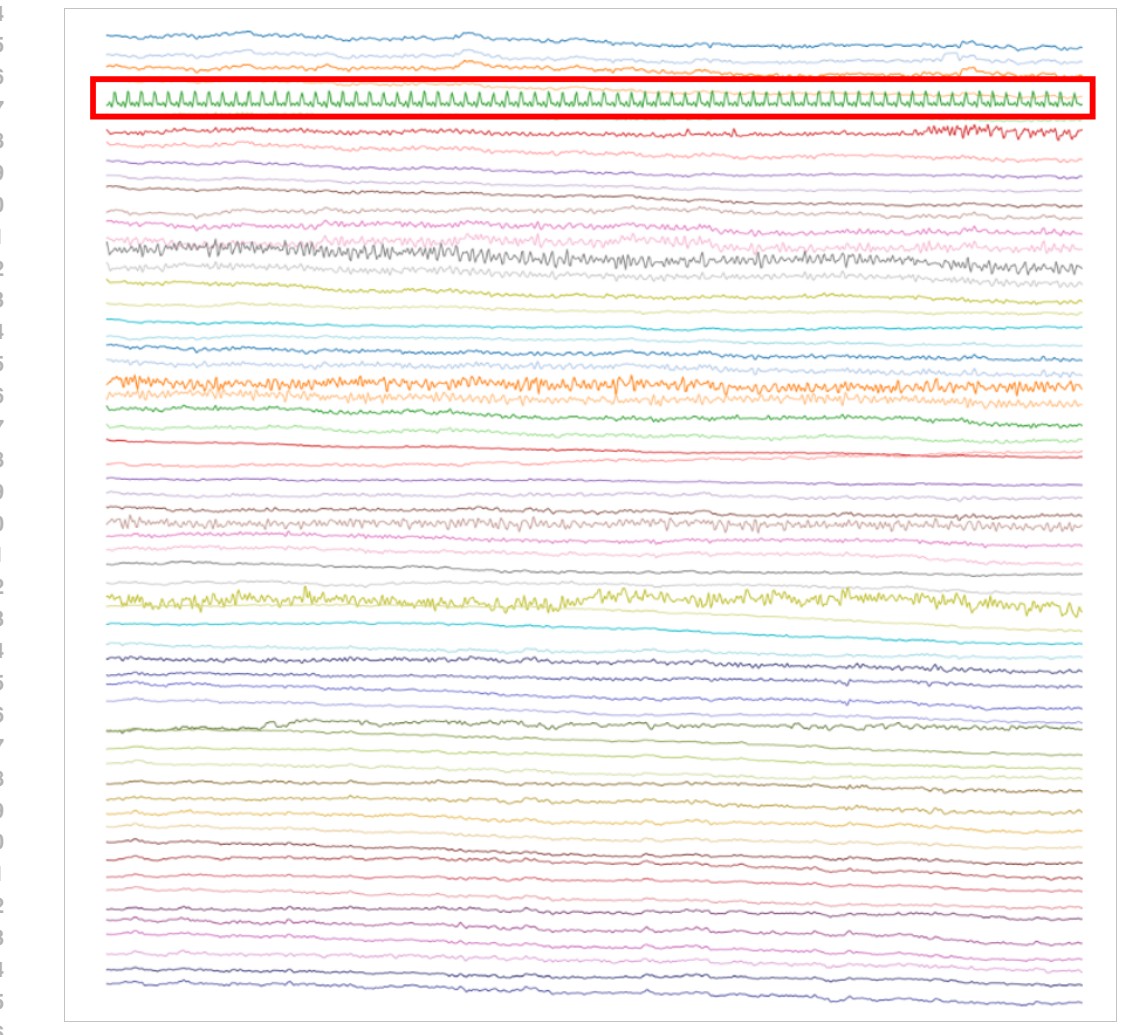

Figure 1: Example of a raw SEED EEG segment with sensor faults highlighted.

## 2.2 MORE EXPERIMENTAL DETAILS

We follow ESTformer and STAD slicing strategies. Preprocessed signals are windowed into fixed lengths: SEED and SEED-IV use non-overlapping $4\,\text{s}$ segments, while Localize-MI retains $-250\,\text{ms}$ to $+10$ ms around each stimulus (260 ms total). We randomly split 80% for training and 20% for testing, yielding $24265 \times 62 \times 800$ train / $6067 \times 62 \times 800$ test samples for SEED; $29199 \times 62 \times 800$ train / $7300 \times 62 \times 800$ test for SEED-IV; and $1914 \times 256 \times 2081$ train / $479 \times 256 \times 2081$ test for Localize-MI. For SEED and SEED-IV we evaluate $2\times$, $4\times$, and $8\times$ super-resolution; for Localize-MI we additionally include $16\times$. As shown in Figure 3, Localize-MI employs a 256-channel intracranial grid, while Figure 4 shows the 62-channel scalp montage used in SEED-IV and SEED.

## 3 MORE SRGDIFF MODEL DETAILS

### 3.1 VARIATIONAL AUTOENCODER

In this paper, Variational Autoencoder (VAE) follows the AutoencoderKL design **?**, comprising a convolutional encoder, a latent distribution (mean and variance) with KL regularization toward $\mathcal{N}(0, I)$, and a decoder with deconvolutions and upsampling. We augment both encoder and decoder

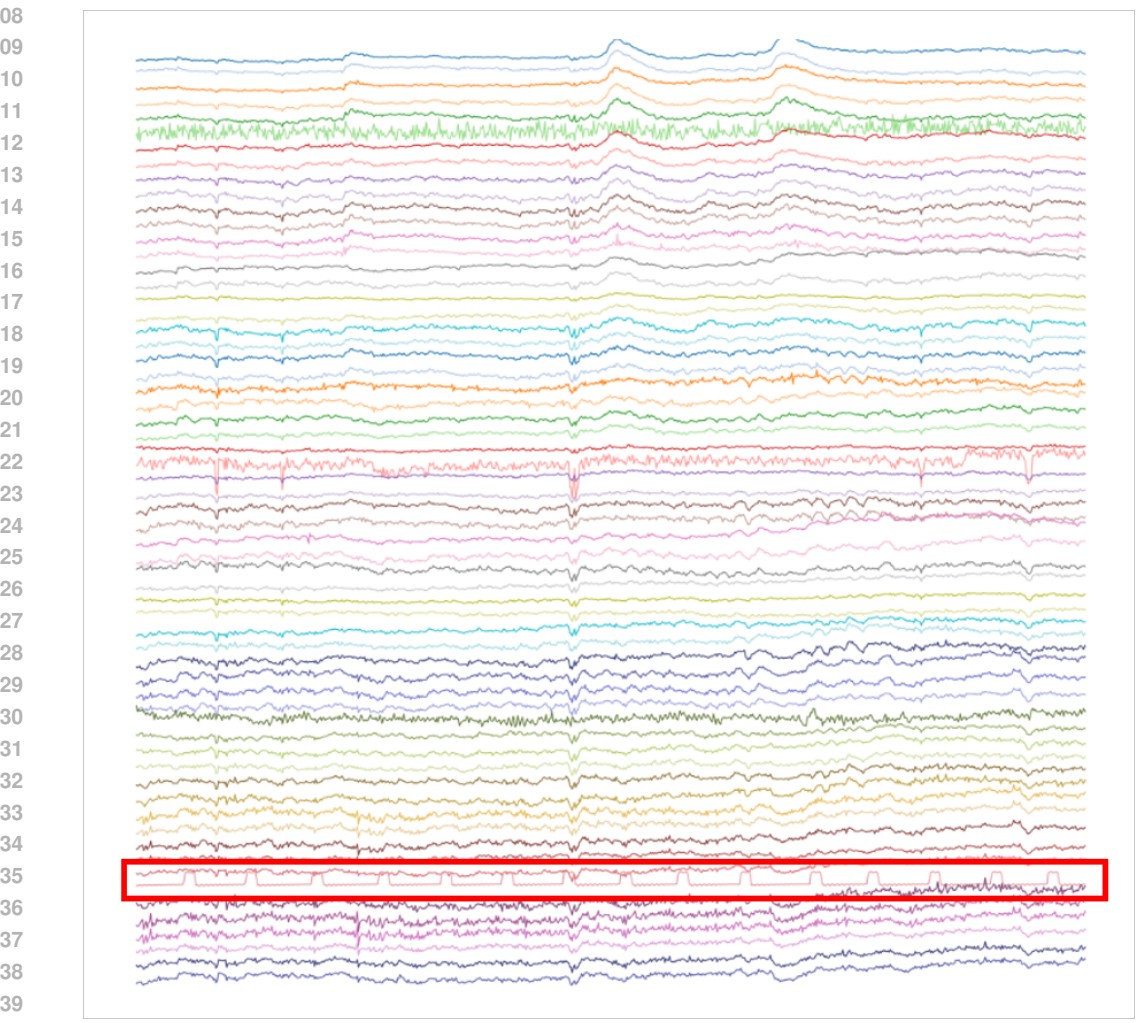

Figure 2: Example of a raw SEED EEG segment with sensor faults highlighted.

with attention layers (multi-head and non-local attention), residual connections, and GroupNorm to capture global EEG features while ensuring stable training and efficient latent representations.

## 3.2 DDIMSCHEDULER

The DDIMScheduler manages noise scheduling and sampling in the forward and reverse diffusion processes. It supports multiple noise prediction types and variance strategies. At each step, it computes the noise coefficient, predicts the denoised sample, clips values for numerical stability, and injects random perturbations to control output diversity.

## 4  PARAMETER STUDY

### 4.1  VAE LATENT SHAPE SELECTION

We found that the latent shape balances reconstruction precision and generalization. Higher dimensions capture more detail but risk overfitting, while lower dimensions blur outputs. We experimented across the three datasets and selected a latent of $32 \times 400$ for SEED/SEED-IV and $64 \times 500$ for Localize-MI, which yielded optimal NMSE, PCC, and SNR. Table 1, Table 2 and Table 3 shows the performance of SRGDiff in different latent shapes.

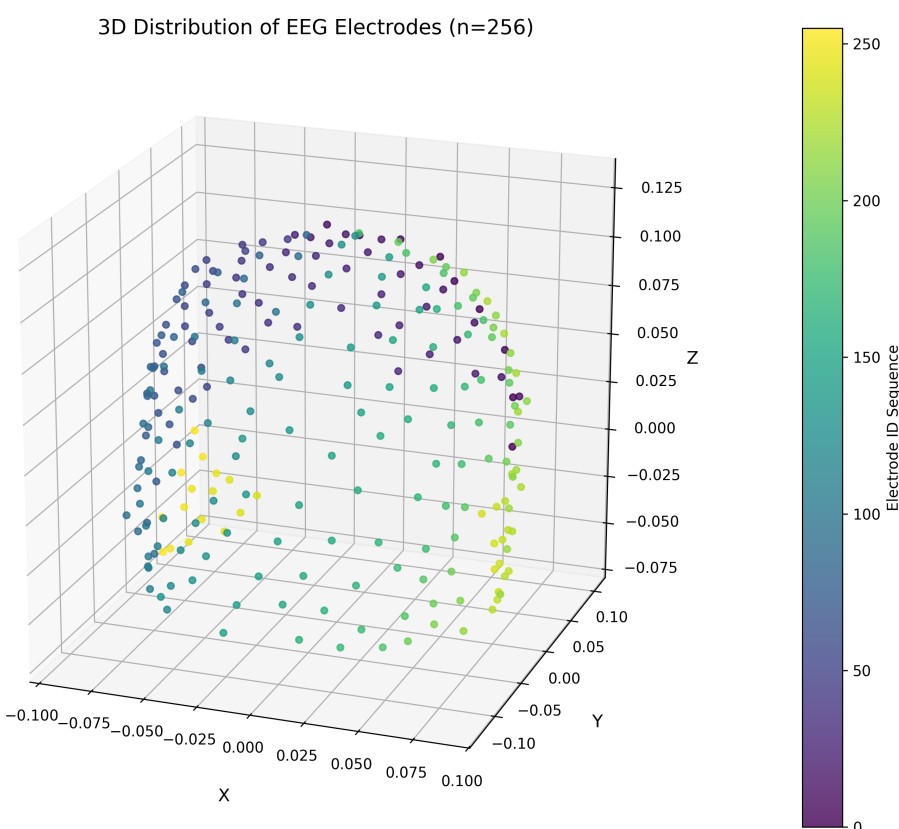

Figure 3: Electrode topology of the Localize-MI dataset (256 intracranial channels).

| Shape | NMSE | PCC | SNR (dB) |
|---|---|---|---|
| $64 \times 400$ | 0.15 | 0.93 | 7.75 |
| **$32 \times 400$** | **0.12** | **0.95** | **8.72** |
| $16 \times 200$ | 0.20 | 0.89 | 6.81 |
| $8 \times 400$ | 0.16 | 0.92 | 7.26 |

Table 1: SEED: VAE latent shape selection results

## 4.2 DIFFUSION HYPERPARAMETERS

### 4.2.1 DIFFUSION SCHEDULES

We compare linear and cosine noise schedules. Linear adds noise at a constant rate but may cause instability at endpoints; cosine offers smoother transitions and better performance for long diffusion chains. We fixed 1000 timesteps with cosine scheduling and evaluated NMSE, PCC, and SNR on the latent reconstructions to choose this setting as shown in Table 4.

### 4.2.2 TRAINING TIMESTEP LENGTHS

We also tested different training timestep lengths (200, 1000, 2000). Larger values introduce stronger noise but make denoising harder; smaller values lack coverage of high-noise regimes. Using cosine scheduling, the results in Table 5 exhibit that 1000 timesteps to be optimal across datasets.

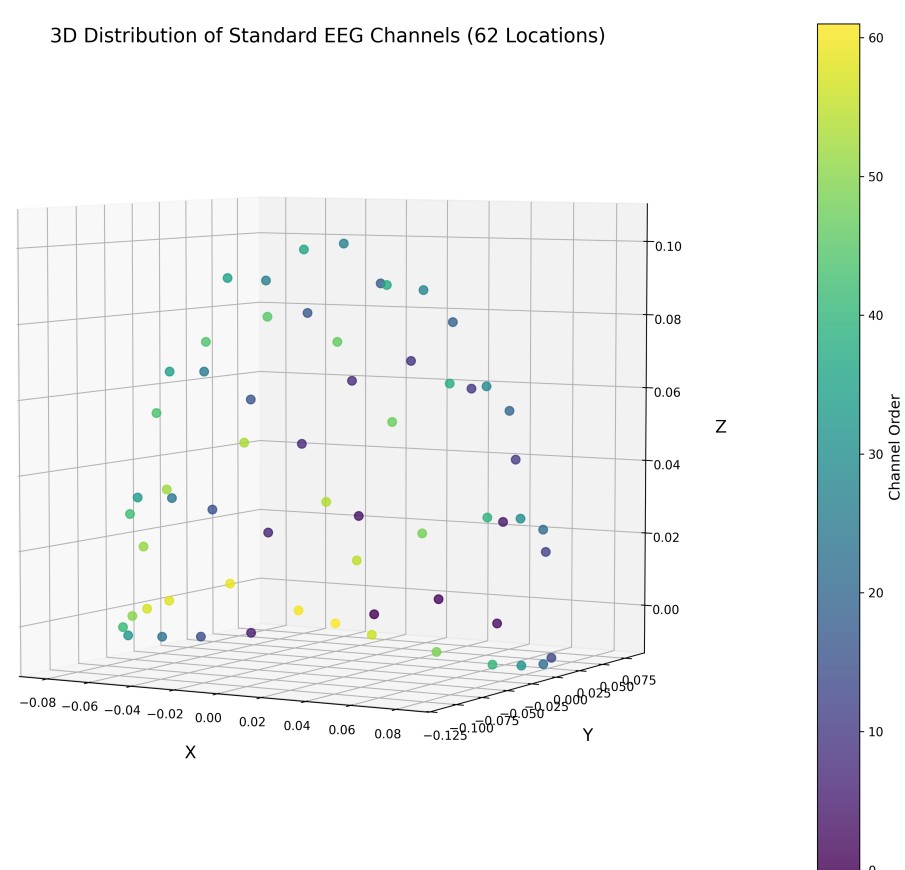

Figure 4: Electrode topology of the SEED-IV and SEED datasets (62 scalp channels).

| Shape | NMSE | PCC | SNR (dB) |
|---|---|---|---|
| $64 \times 400$ | 0.16 | 0.92 | 7.58 |
| **$32 \times 400$** | **0.13** | **0.94** | **8.58** |
| $16 \times 200$ | 0.21 | 0.87 | 6.86 |
| $8 \times 400$ | 0.19 | 0.91 | 7.23 |

Table 2: SEED-IV: VAE latent shape selection results

### 4.2.3 COSINE SCHEDULE OFFSET FACTOR

In the cosine noise schedule for DDIM, the offset factor $s$ adjusts the smoothness and starting point of the noise variance curve to prevent instability from overly small initial noise levels. Concretely, $s$ introduces a phase shift in the cosine function, producing a more gradual noise increase at early timesteps—thereby avoiding abrupt noise jumps—while still covering the full variance range at later steps. Smaller values of $s$ yield gentler initial noise ramp-up, whereas larger $s$ accelerate early noise growth. Table 6 depicts the effect of cosine schedule offset factors on reconstruction quality.

| Shape | NMSE | PCC | SNR (dB) |
|---|---|---|---|
| $128 \times 1000$ | 0.13 | 0.94 | 8.60 |
| **64×500** | **0.09** | **0.96** | **9.01** |
| $32 \times 500$ | 0.15 | 0.92 | 7.61 |
| $32 \times 1000$ | 0.13 | 0.94 | 8.62 |

Table 3: Localize-MI: VAE latent shape selection results

| Dataset | Schedule | NMSE | PCC | SNR (dB) |
|---|---|---|---|---|
| SEED | Linear | 0.42 | 0.71 | 4.18 |
| | **Cosine** | **0.20** | **0.86** | **7.15** |
| SEED-IV | Linear | 0.51 | 0.66 | 4.02 |
| | **Cosine** | **0.19** | **0.88** | **7.24** |
| Localize-MI | Linear | 0.14 | 0.93 | 8.39 |
| | **Cosine** | **0.11** | **0.95** | **8.88** |

Table 4: Comparison of noise schedules on three datasets (NMSE, PCC, SNR).

## 5 DOWNSTREAM TASK

### 5.1 CLASSIFICATION FEATURE EXTRACTION

#### 5.1.1 DIFFERENTIAL ENTROPY FEATURE

Raw EEG at 1000 Hz is downsampled to 200 Hz, band-pass filtered (1–50 Hz) with a 6th-order Butterworth filter, and segmented into non-overlapping 1 s windows (200 samples). Each window is transformed by STFT (Hanning window, 200-point length, 256-point FFT). We compute band-specific power $E$ for $\delta$ (1–3 Hz), $\theta$ (4–7 Hz), $\alpha$ (8–13 Hz), $\beta$ (14–30 Hz), and $\gamma$ (31–50 Hz) , normalize by the number of bins $N$, and define differential entropy (DE) feature as $\log(E/N)$ with a small constant added for numerical stability.

#### 5.1.2 POWER SPECTRAL DENSITY FEATURE

Power spectral density (PSD) feature features use the same STFT pipeline but report the mean squared magnitude (average power) in each band.

### 5.2 RANDOM FOREST CLASSIFIER

For emotion classification on SEED and SEED-IV and epileptic detection on Localize-MI, we employ a random forest with 100 trees. This set of hyperparameters balances nonlinearity modeling with computational efficiency, yielding robust performance on high-dimensional EEG features.

## 6 SUPPLEMENTAL ABLATION RESULTS

Figure 5 presents additional ablation studies on SEED-IV and Localize-MI datasets, reporting NMSE, PCC, and SNR for the baseline LDM, LDM+SMM, LDM+RDM, and the full SRGDiff across various upsampling scales. These plots further illustrate the individual and combined contributions of our two modules to reconstruction quality.

## 7 RECONSTRUCTION VISUALIZATION

Figure 6 through Figure 8 illustrate qualitative reconstructions on the three datasets. For each, we plot a single representative channel over time, comparing the ground-truth high-density EEG

| Dataset | Steps | NMSE | PCC | SNR (dB) |
|---|---|---|---|---|
| SEED | 200 | 0.32 | 0.76 | 5.95 |
| | **1000** | **0.20** | **0.86** | **7.15** |
| | 2000 | 0.29 | 0.78 | 6.19 |
| SEED-IV | 200 | 0.36 | 0.75 | 5.93 |
| | **1000** | **0.19** | **0.88** | **7.24** |
| | 2000 | 0.31 | 0.76 | 6.11 |
| Localize-MI | 200 | 0.15 | 0.92 | 8.37 |
| | 1000 | 0.11 | 0.95 | 8.88 |
| | **2000** | **0.11** | **0.95** | **8.94** |

Table 5: Impact of training timesteps on reconstruction quality

| Dataset | $s$ | NMSE | PCC | SNR (dB) |
|---|---|---|---|---|
| SEED | **0.005** | **0.20** | **0.86** | **7.15** |
| | 0.010 | 0.24 | 0.84 | 7.02 |
| | 0.025 | 0.26 | 0.83 | 6.93 |
| SEED-IV | 0.005 | 0.20 | 0.86 | 7.17 |
| | **0.010** | **0.19** | **0.88** | **7.24** |
| | 0.025 | 0.20 | 0.85 | 7.11 |
| Localize-MI | **0.005** | **0.11** | **0.95** | **8.94** |
| | 0.010 | 0.15 | 0.93 | 8.41 |
| | 0.025 | 0.18 | 0.91 | 8.23 |

Table 6: Effect of cosine schedule offset factor $s$ on reconstruction quality

(black) against STAD (blue), ESTformer (red), and SRGDiff (green). These overlays demonstrate SRGDiff's closer alignment with the true waveform across diverse temporal patterns.

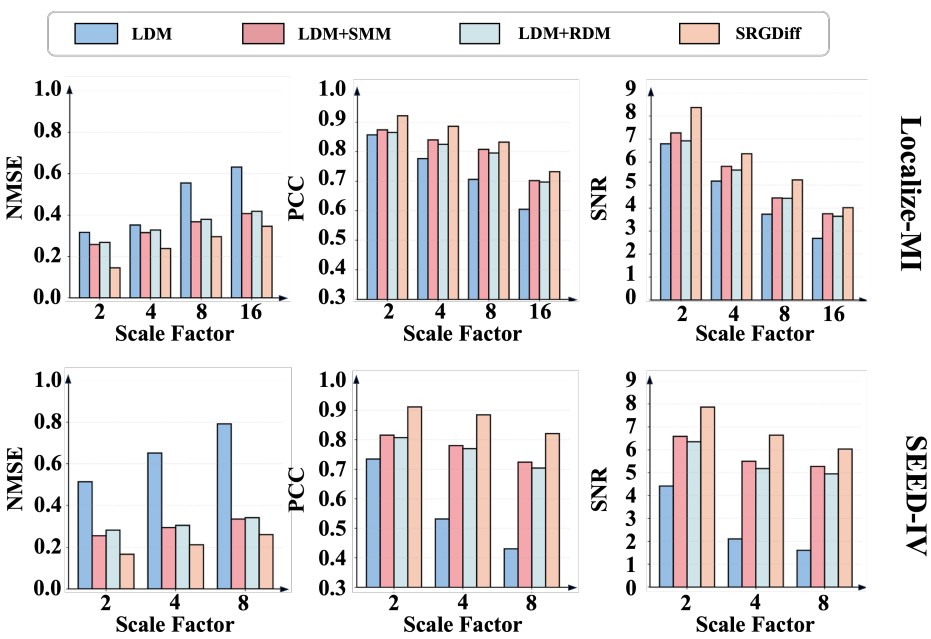

Figure 5: Ablation results on SEED-IV and Localize-MI: comparison of NMSE, PCC, and SNR for LDM, LDM+SMM, LDM+RDM, and SRGDiff at 2×, 4×, and 8× upsampling and 2×, 4×, 8×, and 16× upsampling, respectively.

Table 7: Runtime (ms) of different methods on SEED, SEED-IV, and Localize-MI datasets.

| Method | SEED | SEED-IV | MI |
|---|---|---|---|
| SasDim | 265.4 | 269.5 | 374.4 |
| SADI | 329.0 | 324.1 | 428.7 |
| RDPI | 318.1 | 315.9 | 421.2 |
| DDPMEEG | 549.3 | 558.2 | 841.7 |
| ESTformer | 3.51 | 3.55 | 5.03 |
| STAD | 232.6 | 227.8 | 385.9 |
| Ours | 63.0 | 62.1 | 92.5 |

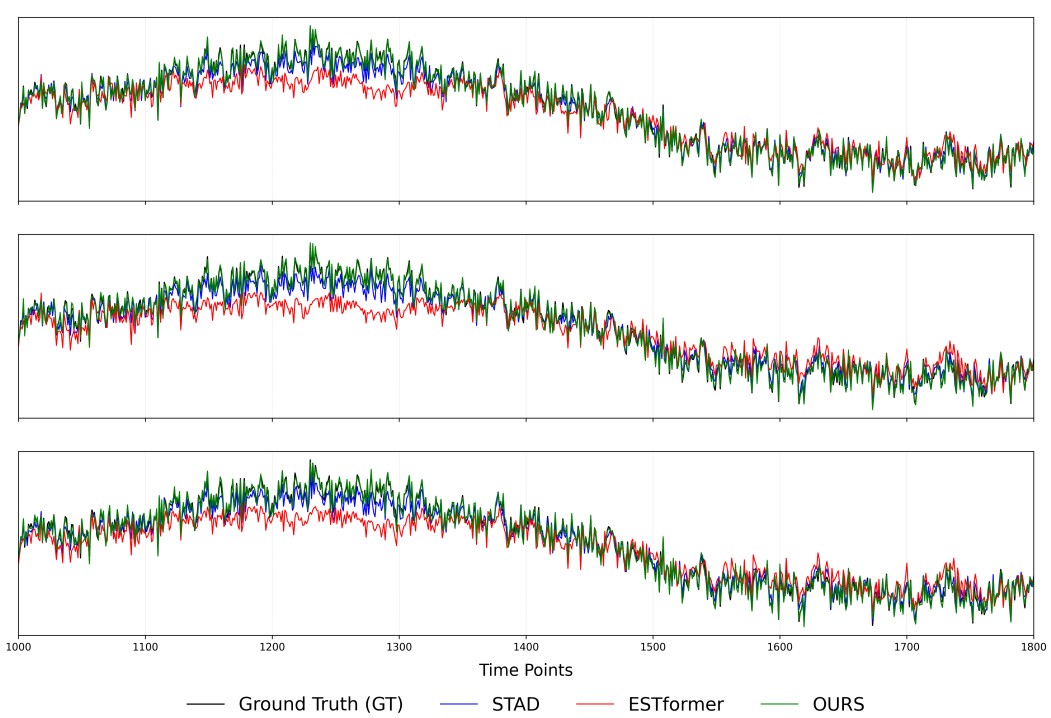

Figure 6: Reconstruction of a representative channel on Localize-MI (motor imagery): ground truth (black), STAD (blue), ESTformer (red), and SRGDiff (green).

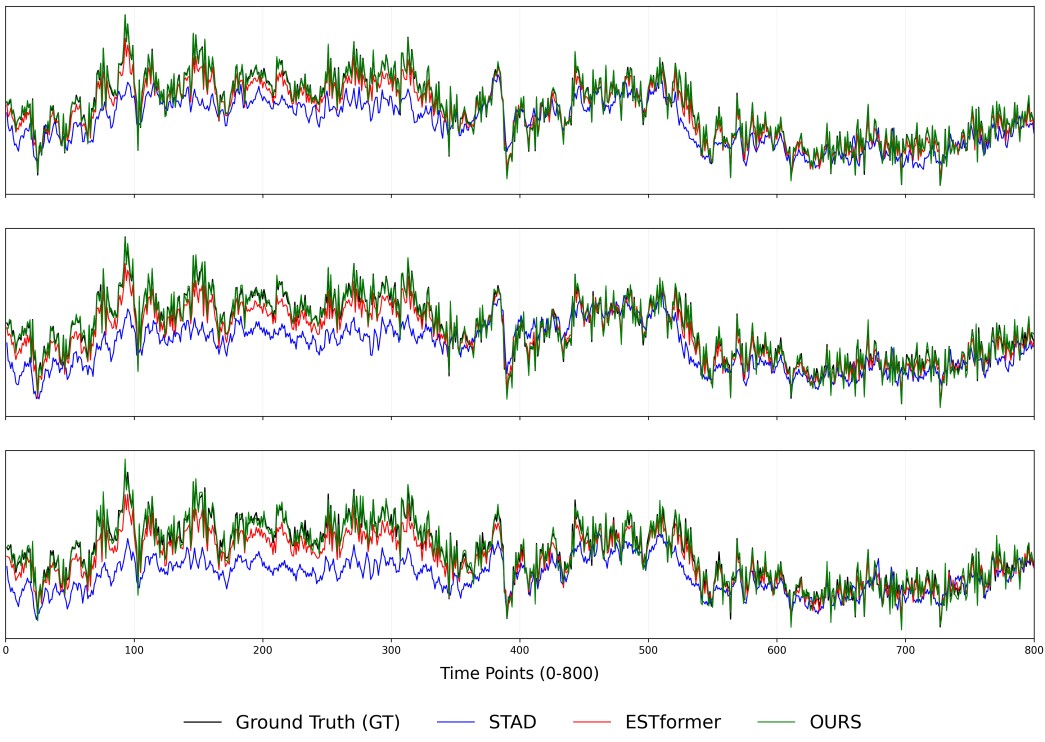

Figure 7: Reconstruction of a representative channel on SEED-IV (emotion recognition): ground truth (black), STAD (blue), ESTformer (red), and SRGDiff (green).

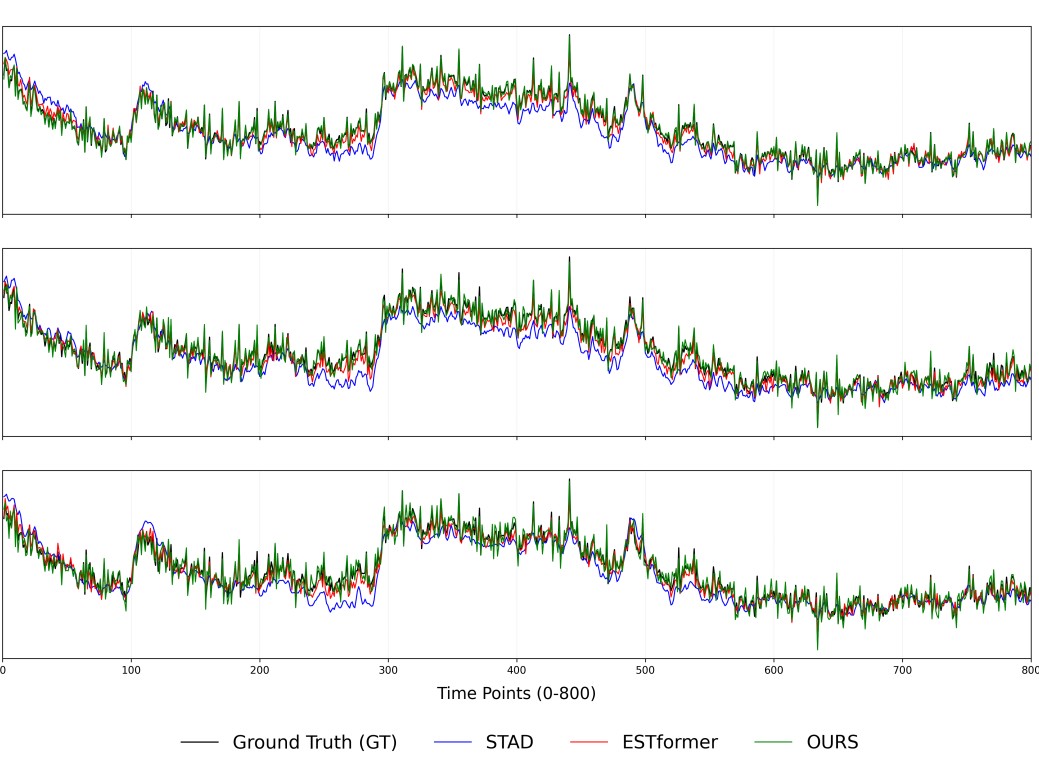

Figure 8: Reconstruction of a representative channel on SEED (emotion recognition): ground truth (black), STAD (blue), ESTformer (red), and SRGDiff (green).