# OpenReview forum: "Step-Aware Residual-Guided Diffusion for EEG Spatial Super-Resolution"
_ICLR.cc/2026/Conference — ICLR 2026 Poster_

### Official Review · Reviewer_pkZv · 2025-10-20

**Soundness:** 2
**Presentation:** 4
**Contribution:** 3
**Rating:** 6
**Confidence:** 4

**Summary:**

This paper tackles the problem of EEG spatial super-resolution (SR), which aims to reconstruct high-density (HD) EEG signals from sparse, low-density (LD) measurements. The authors argue that existing methods, including recent diffusion models, use a "static guidance" strategy (like concatenation) that leads to a trade-off between fidelity to the HD signal distribution and consistency with the LD input .
To solve this, they propose SRGDiff, a latent diffusion model that uses a novel "dynamic conditional generation" framework. The core idea is to learn a dynamic residual from the LD input, which acts as a step-aware directional correction during the reverse diffusion (denoising) process. This is implemented via two new modules: a Residual Direction Module (RDM) that predicts this step-wise residual and a Step-Aware Modulation Module (SMM) that applies step-dependent affine modulation to control the guidance strength. The authors evaluate their method on three datasets (SEED, SEED-IV, Localize-MI) using a comprehensive three-level protocol (signal, feature, and downstream) , demonstrating state-of-the-art results that significantly outperform strong baselines.

**Strengths:**

The core concept of "dynamic residual guidance" is a clever and novel departure from standard static conditioning in diffusion models.
The paper demonstrates massive, consistent performance gains over a suite of strong, recent baselines (e.g., ESTformer, STAD) across three datasets . A 50% NMSE reduction on SEED is a remarkable improvement.
The three-level evaluation protocol (signal-level NMSE/PCC, feature-level EEG-FID, and downstream classification accuracy) is a model of rigor and provides a holistic and convincing picture of the generated signals' quality and utility .
Despite its architectural complexity, the authors provide evidence (Table 10) that their method is significantly faster than other diffusion-based approaches, making it more practical.

**Weaknesses:**

The authors motivate their work by criticizing "static guidance"  but then fail to include a static guidance model (e.g., LDM with feature concatenation) in their ablation study (Figure 5). The comparison against a seemingly unconditional LDM baseline is a feeble argument. This omission makes it impossible to validate the paper's central claim: that the proposed dynamic RDM and SMM modules are superior to the standard, simpler conditioning method they are designed to replace.

**Questions:**

In your ablation study (Figure 5), is the "LDM" baseline an unconditional model? If so, why did you not compare against the primary target of your criticism: a standard statically-guided LDM (e.g., using feature concatenation), which would serve as a proper baseline to evaluate the utility of your RDM and SMM modules?

The performance gains in your ablation (e.g., NMSE 0.86 for LDM vs. 0.34 for SRGDiff at 8x on SEED)  are far larger than the gains over the SOTA conditional model STAD (Table 2). This strongly suggests your LDM baseline is unconditional, making the ablation misleading. Can you provide a direct comparison between SRGDiff and an "LDM + Static Concatenation" baseline?

Your runtime in Table 10 is impressively fast. Is this because the RDM and SMM are very lightweight? And does the static-guidance-by-concatenation baseline (which I am asking for) run even faster?

---

> ### Author Response · Authors · 2025-11-20
> **We thank Reviewer pkZv for the effort in reviewing our work and providing supportive comments. We address each concern or question below.**
>
> > W1: The authors motivate their work by criticizing "static guidance" but then fail to include a static guidance model (e.g., LDM with feature concatenation) in their ablation study (Figure 5). The comparison against a seemingly unconditional LDM baseline is a feeble argument. This omission makes it impossible to validate the paper's central claim: that the proposed dynamic RDM and SMM modules are superior to the standard, simpler conditioning method they are designed to replace.
> >
> > Q1: In your ablation study (Figure 5), is the "LDM" baseline an unconditional model? If so, why did you not compare against the primary target of your criticism: a standard statically-guided LDM (e.g., using feature concatenation), which would serve as a proper baseline to evaluate the utility of your RDM and SMM modules?
>
> **Response.** Thank you for pointing this out and for flagging the terminology issue. We clarify that our “LDM” baseline in Fig. 5 is not truly unconditional but represents "LDM+static guidance". In EEG SR, HD reconstruction always requires the LD observation as input. In the ablation, the LD EEG is first mapped to the latent feature space via the pretrained VAE encoder and then added to the diffusion feature as "static guidance". In other words, the ablation model "LDM" already uses a static guidance, but our description simply called it “LDM”, which understandably gave the impression of an unconditional model. In the revised version, we rename this baseline to “LDM+LD” to make it explicit that low-density EEG is used as the conditioning signal in the ablation.
>
> > Q2: The performance gains in your ablation (e.g., NMSE 0.86 for LDM vs. 0.34 for SRGDiff at 8x on SEED) are far larger than the gains over the SOTA conditional model STAD (Table 2). This strongly suggests your LDM baseline is unconditional, making the ablation misleading. Can you provide a direct comparison between SRGDiff and an "LDM + Static Concatenation" baseline?
>
> **Response.** As clarified in our response to Q1, the “LDM” baseline in the ablation is a *conditionally guided* latent diffusion model. After re-checking the numbers, we also found that “the gains over LDM are far larger than the gains over STAD” is not accurate.
>
> To make this explicit, we summarize the NMSE results on SEED (2×/4×/8×) by extracting the numbers from the main tables. As the table shows, the static-guided LDM baseline is comparable to STAD: it is slightly lower at $2\times$ but slightly higher at $4\times$ and $8\times$. We attribute this moderate advantage to the fact that our LDM+LD baseline still benefits from a two-stage training strategy. When taking LD inputs, the signals are first passed through the pretrained VAE encoder; at this stage, the guidance features already inherit some high-resolution semantic structure learned during VAE pretraining. As a result, even though the conditioning is relatively simple, the latent diffusion model can still exploit these semantics and remains competitive with STAD, especially at the more challenging $8\times$ SR factor.
>
>
> | Model                     | Metric | 2×     | 4×     | 8×     |
> |---------------------------|--------|--------|--------|--------|
> | STAD                      | NMSE   | 0.4319 | 0.6913 | 0.8617 |
> | LDM + LD                  | NMSE   | 0.5134 | 0.6433 | 0.7987 |
> | **SRGDiff**               | NMSE   | **0.1632** | **0.2977** | **0.3494** |
>
>
> > Q3: Your runtime in Table 10 is impressively fast. Is this because the RDM and SMM are very lightweight? And does the static-guidance-by-concatenation baseline (which I am asking for) run even faster?
>
> **Response.** We appreciate the positive feedback on runtime. RDM and SMM are indeed lightweight modules, but the main reason for the fast inference is the overall design of SRGDiff rather than these two blocks alone. Specifically, SRGDiff operates entirely in the latent HD space, and at inference time we use a fixed small number of DDIM sampling steps, so each test sample runs the same short denoising chain regardless of the dataset or SR factor.
>
> To quantify the overhead of RDM and SMM, we measure the average wall-clock inference time (milliseconds per 4s input window, batch size 1) on the same GPU for the latent diffusion backbone (LDM) and its variants. The results show that adding SMM or RDM on top of LDM increases runtime only marginally, and the full SRGDiff model remains within single-digit percentage overhead compared to the LDM backbone. This confirms that the RDM/SMM conditioning is computationally light, and that most of the cost still comes from the shared VAE encoder/decoder and U-Net backbone, rather than our guidance modules.
>
> | Method   | SEED | SEED-IV | Localize-MI |
> |----------|:----:|:-------:|:-----------:|
> | LDM+LD      | 60.8 | 60.3    | 85.3        |
> | LDM+SMM  | 61.4 | 60.8    | 86.9        |
> | LDM+RDM  | 61.9 | 61.3    | 87.2        |
> | SRGDiff  | 63.0 | 62.1    | 92.5        |

---

> > ### Author Response · Authors · 2025-11-26
> > **Summary of all addressed points**
> >
> > Thank you again for your detailed feedback and for pointing out ambiguities in our ablation setup and runtime discussion; this helped us substantially clarify the role of the LDM baseline and our conditioning modules.
> >
> > In the revised version, we have addressed your main comments as follows
> >
> > * The "LDM" baseline in the ablation is not unconditional but already uses static guidance from LD EEG in the latent space. To avoid confusion, we now explicitly rename this baseline to “LDM+LD” throughout the paper.
> > * An additional explicit comparison between SRGDiff, STAD, and the LDM+LD baseline by summarizing their NMSE on SEED for 2×/4×/8× SR.
> > * A new runtime analysis have been added to quantify the computational overhead of our conditioning modules, demonstrating the light weight of the RDM and SMM module.
> >
> > We hope these clarifications resolve the concerns about our ablation design and runtime claims, and we would be very grateful if you could kindly reassess your overall rating in light of the revised manuscript.

---

### Official Review · Reviewer_cWrT · 2025-10-25

**Soundness:** 2
**Presentation:** 4
**Contribution:** 3
**Rating:** 4
**Confidence:** 4

**Summary:**

This paper addresses the emerging problem of EEG spatial super-resolution, a relatively new and challenging topic in the EEG community. While super-resolution has been extensively studied in computer vision, only a few works have recently explored this concept for EEG, primarily in IEEE journals rather than mainstream neuroscience venues. The authors adapt state-of-the-art deep learning tools, such as transformers and generative diffusion models, to reconstruct high-density EEG from low-density recordings. This effort bridges the methodological gap between biomedical signal processing and modern deep generative modeling, which is commendable. However, several conceptual and technical issues require deeper clarification and refinement to make the work truly convincing from both a signal processing and machine learning standpoint.

**Strengths:**

The paper introduces EEG spatial super-resolution to the broader ML community, addressing a domain problem that remains underexplored.

The use of deep generative modeling (transformer/diffusion) provides a fresh analytical perspective and helps re-interpret a traditionally biomedical task in a form understandable to the ICLR community.

The overall framework and evaluation pipeline are consistent with prior EEG SR baselines (e.g., ESTformer, STAD), allowing fair comparison and reproducibility.

**Weaknesses:**

1. The authors continue using NMSE, PCC, and SNR as the main evaluation metrics. These are inherited from previous EEG works but lack a theoretical justification for super-resolution tasks in the ML setting. Given the ICLR context, this presents an opportunity to re-examine whether these metrics are conceptually valid critically. For instance, does high-density (HD) EEG necessarily yield higher SNR values than low-density EEG? If not, why do the reported gains occur?

2. EEG signals vary significantly across sessions and subjects. Without explicitly addressing this factor, through normalization, adaptation, or domain alignment, the model’s reported performance might not generalize. The paper does not mention whether such variability was handled or controlled.

3. When compared to recent baselines such as STAD and ESTformer, the proposed method shows similar results across several datasets and metrics. The manuscript lacks a clear and intuitive explanation of how the proposed architectural modifications lead to consistent improvements beyond fine-tuned engineering choices.

4. Figure 4 appears to showcase selected examples. Quantitative evidence (e.g., statistical comparisons or full-dataset averages) should be provided to demonstrate consistent superiority.

**Questions:**

1. Why did you decide to continue using NMSE, PCC, and SNR? Could you justify why these metrics are still meaningful for evaluating generative EEG models, especially when HD EEG may not inherently outperform LD EEG on such metrics?

2. How do you handle session- and subject-level variability? Were normalization or adaptation techniques applied to ensure robustness?

3. Can you provide a more intuitive explanation for the specific architectural innovations that differentiate this method from STAD or ESTformer?

4. Could you include full quantitative comparisons corresponding to Figure 4 to rule out visual selection bias?

---

> ### Author Response · Authors · 2025-11-20
> **We thank Reviewer cWrT for the effort in reviewing our work and providing supportive comments. We address each concern or question below.**
>
> > W1: The authors continue using NMSE, PCC, and SNR as the main evaluation metrics. These are inherited from previous EEG works but lack a theoretical justification for super-resolution tasks in the ML setting. Given the ICLR context, this presents an opportunity to re-examine whether these metrics are conceptually valid critically. For instance, does high-density (HD) EEG necessarily yield higher SNR values than low-density EEG? If not, why do the reported gains occur?
> >
> > Q1: Why did you decide to continue using NMSE, PCC, and SNR? Could you justify why these metrics are still meaningful for evaluating generative EEG models, especially when HD EEG may not inherently outperform LD EEG on such metrics?
>
> **Response.** Thank you for this insightful comment. Our use of NMSE, PCC, and SNR follows a long line in EEG and other classical signal processing on super-resolution and reconstruction [1-6]. In classical image SR problems [1-2], pointwise distortion measures such as MSE/PSNR and structural similarity are standard for quantifying how well a reconstructed high-resolution signal matches a ground-truth reference. Existing EEG spatial SR methods [4-6] have adopted the same family of metrics for HD-from-LD reconstruction, and we follow this established practice.
>
> From an ML perspective, EEG SR can be viewed as learning the conditional distribution $ p(X^{H}\mid X^{L}) $, where $ X^{H} $ is HD EEG and $ X^{L} $ is its LD observation. Under a squared-error loss, the natural point estimator is the conditional mean $ \hat X = \mathbb{E}[X^{H}\mid X^{L}] $. Our normalized MSE
> $$
> \mathrm{NMSE} = \frac{\lVert \hat X - X^{H} \rVert_2^2}{\lVert X^{H} \rVert_2^2}
> $$
> directly measures how close the learned estimator is to this MMSE-optimal reconstructor, normalized for cross-subject / cross-dataset comparability. This is analogous to the standard use of MSE/PSNR in image SR.
>
> However, many EEG models are sensitive not only to amplitudes but also to temporal covariance structure. We therefore also report the Pearson correlation coefficient (PCC),
> $$
> \mathrm{PCC} = \frac{\langle \hat X, X^{H} \rangle}{\lVert \hat X \rVert_2 ,\lVert X^{H} \rVert_2},
> $$
> which checks whether the reconstructed signal preserves the normalized second-order structure of the HD reference that downstream BCI models actually rely on.
>
> The SNR in our tables is **reconstruction SNR** rather than the intrinsic hardware SNR of HD vs. LD caps:
> $$
> \mathrm{SNR} = 10\log_{10} \frac{\lVert X^{H} \rVert_2^2}{\lVert \hat X - X^{H} \rVert_2^2}.
> $$
>
> Thus, a better reconstruction will necessarily yield higher *reconstruction SNR* because it reduces the error energy with respect to the HD target. This explains why SNR gains appear in our results and why they remain meaningful for reference-based generative SR.
>
> In addition to NMSE/PCC/SNR at the signal level, we also report (i) an EEG-FID–style metric computed on a frozen EEG encoder to assess distribution-level faithfulness in feature space, and (ii) task-level utility on downstream BCI and emotion recognition tasks.
>
> Reference
>
> [1] Ledig C, Theis L, Huszár F, et al. Photo-realistic single image super-resolution using a generative adversarial network[C]//Proceedings of the IEEE conference on computer vision and pattern recognition. 2017: 4681-4690.
>
> [2] Mardani M, Gong E, Cheng J Y, et al. Deep generative adversarial neural networks for compressive sensing MRI[J]. IEEE transactions on medical imaging, 2018, 38(1): 167-179.
>
> [3] Lai Y, Chen J, Zhao Q, et al. DiffuSETS: 12-Lead ECG generation conditioned on clinical text reports and patient-specific information[J]. Patterns, 2025.
>
> [4] Li D, Zeng Z, Wang Z, et al. ESTformer: Transformer utilising spatiotemporal dependencies for electroencephalogram super-resolution[J]. Knowledge-Based Systems, 2025, 317: 113345.
>
> [5] Tang Y, Chen D, Liu H, et al. Deep EEG superresolution via correlating brain structural and functional connectivities[J]. IEEE Transactions on Cybernetics, 2022, 53(7): 4410-4422.
>
> [6] Wang S, Zhou T, Shen Y, et al. Generative AI enables EEG super-resolution via spatio-temporal adaptive diffusion learning[J]. IEEE Transactions on Consumer Electronics, 2025.

---

> > ### Author Response · Authors · 2025-11-20
> > **Continued response (2/3)**
> >
> > > W2: EEG signals vary significantly across sessions and subjects. Without explicitly addressing this factor, through normalization, adaptation, or domain alignment, the model’s reported performance might not generalize. The paper does not mention whether such variability was handled or controlled.
> > >
> > > Q2: How do you handle session- and subject-level variability? Were normalization or adaptation techniques applied to ensure robustness?
> >
> > **Response.** We agree that session- and subject-level variability is a major challenge for EEG and should be made more explicit in the paper. We have added a dedicated cross-subject / cross-session generalization analysis. Concretely, for the cross-session setting on SEED, we use all subjects and perform three experiments where we train on two sessions and test on the remaining one, and report averages over all subjects. For the cross-subject setting, we train on subjects 1–12 and evaluate on held-out subjects 13–15.
> >
> > We observe that, in the cross-session setting, SRGDiff maintains strong performance at $2\times$ and $4\times$ SR with only modest degradation relative to the within-session setup, while the drop becomes more pronounced at $8\times$. In contrast, the cross-subject setting is substantially more challenging and leads to larger performance degradation across all SR factors. Nevertheless, across both cross-session and cross-subject settings and all SR factors, SRGDiff consistently maintains a clear margin over strong baselines in terms of NMSE, PCC, and SNR. Further details are reported in the revised manuscript in Appendix H.1, Tables 15 and 16.
> >
> > | model     | metrics | Cross_subject      |      |      | Cross_session      |      |      |
> > |-----------|--------|--------------------------|------|------|--------------------------|------|------|
> > |           |        | $2\times$                        | $4\times$    | $8\times$    | $2\times$                        | $4\times$    | $8\times$    |
> > | ESTformer | NMSE↓   | 0.4411±0.004             | 0.4729±0.006 | 0.5633±0.008 | 0.3624±0.0038          | 0.4029±0.0079 | 0.5129±0.0072 |
> > |           | PCC↑    | 0.7393±0.009             | 0.7189±0.007 | 0.6515±0.009 | 0.7924±0.0094          | 0.7742±0.0068 | 0.6954±0.0094 |
> > |           | SNR↑    | 3.9516±0.031             | 3.5414±0.004 | 2.7279±0.039 | 4.8753±0.0342          | 4.4097±0.0471 | 3.1375±0.0429 |
> > | STAD      | NMSE↓  | 0.5537±0.004             | 0.7325±0.002 | 0.9267±0.008 | 0.5617±0.003           | 0.7675±0.002  | 0.9376±0.007  |
> > |           | PCC↑    | 0.6224±0.003             | 0.4587±0.002 | 0.2959±0.004 | 0.6373±0.003           | 0.4397±0.002  | 0.2791±0.004  |
> > |           | SNR↑    | 3.3872±0.097             | 1.2449±0.049 | 0.7276±0.139 | 3.1794±0.089           | 1.1297±0.042  | 0.7168±0.122  |
> > | SRGDiff   | NMSE↓   | 0.2675±0.003             | 0.3829±0.005 | 0.4512±0.005 | 0.2480±0.003           | 0.3529±0.004  | 0.4127±0.005  |
> > |           | PCC↑    | 0.8023±0.004             | 0.7232±0.004 | 0.6902±0.004 | 0.8508±0.004           | 0.7932±0.003  | 0.7702±0.007  |
> > |           | SNR↑    | 5.6913±0.067             | 4.5657±0.055 | 4.1189±0.084 | 6.1473±0.093           | 4.1857±0.052  | 3.8189±0.034  |
> >
> > > W3: When compared to recent baselines such as STAD and ESTformer, the proposed method shows similar results across several datasets and metrics. The manuscript lacks a clear and intuitive explanation of how the proposed architectural modifications lead to consistent improvements beyond fine-tuned engineering choices.
> > >
> > > Q3: Can you provide a more intuitive explanation for the specific architectural innovations that differentiate this method from STAD or ESTformer?
> >
> > **Response.** Thank you for asking for a more intuitive explanation. Our original presentation may have made the differences look smaller than they are. We now summarize the relative gains over the baseline ESTformer and STAD, SRGDiff reduces NMSE by roughly 20–50\%, improves SNR by more than 20\% (up to 75\%), and decreases frequency-domain MAE by about 46\% across the three datasets. Intuitively, these improvements come from replacing a single-shot LD$\to$HD mapping with a multi-step, uncertainty-aware residual refinement that better exploits the HD prior under severe under-sampling. SRGDiff changes how LD information is injected into the model and how the HD signal is reconstructed over time.
> >
> > | Dataset     | NMSE↓ | PCC↑ | SNR↑ | Freq-MAE↓  | Cls. Acc↑ |
> > |------------|-------------------------:|------------------:|------:|-------:|-----------------------:|
> > | SEED       | 26.9% | 6.9%              | 29.3%             | 46.1%     | 3.4%                   |
> > | SEED-IV    | 49.0%  | 14.0%             | 75.4%             | 46.7%      | 0.8%                   |
> > | Localize-MI| 19.5%      | 9.3%    | 21.1%             | 46.0% | 5.7%                   |
> > | **Overall mean** | **31.8%**    | **10.1%**  | **41.9%**         | **46.3%**                   | **3.3%**               |

---

> > > ### Author Response · Authors · 2025-11-20
> > > **Continued response (3/3)**
> > >
> > > > W4: Figure 4 appears to showcase selected examples. Quantitative evidence (e.g., statistical comparisons or full-dataset averages) should be provided to demonstrate consistent superiority.
> > > >
> > > > Q4: Could you include full quantitative comparisons corresponding to Figure 4 to rule out visual selection bias?
> > >
> > > **Response.** Thank you for bringing this up. Figure 4 indeed shows a small number of representative EEG segments and topomaps for qualitative visualization. This style of visualization is widely used in deep-learning–based EEG representation and super-resolution works, where authors typically display either a few representative trials or averaged topomaps to illustrate spectral–spatial patterns [1–4]. Our intention with Figure 4 is similarly to provide an intuitive, visual comparison of typical patterns, not to claim superiority based on a single hand-picked example.
> > >
> > > To directly address your concern about potential selection bias, we additionally provide quantitative, full-dataset comparisons as follows. We introduced a new quantitative metric: the frequency-domain MAE between reconstructed and real HD topomaps, computed as described above. We measure frequency-domain MAE by first transforming both reconstructed and real HD EEG into the frequency domain, aggregating power within standard EEG bands, and interpolating the band power onto 2D scalp topomaps using electrode coordinates. We then compute the pixel-wise mean absolute error between the reconstructed and real topomaps, averaged over all frequency bands and test samples. A lower value indicates that the model better preserves the spectral content and its spatial distribution over the scalp.
> > >
> > > The results are shown in the table below. SRGDiff consistently achieves the lowest frequency MAE across all datasets and SR scales, and the gap over baselines becomes even larger in this metric. The full definition of this metric and complete quantitative results are provided in the supplemental material, Subsection F.1.
> > >
> > > Reference
> > >
> > > [1] Wang S, Zhou T, Shen Y, et al. Generative AI enables EEG super-resolution via spatio-temporal adaptive diffusion learning[J]. IEEE Transactions on Consumer Electronics, 2025.
> > >
> > > [2] Li D, Zeng Z, Wang Z, et al. ESTformer: Transformer utilising spatiotemporal dependencies for electroencephalogram super-resolution[J]. Knowledge-Based Systems, 2025, 317: 113345.
> > >
> > > [3] Yin H L, Zheng W L, Lu B L. STAR: A Spatial-Temporal Autoencoder for EEG Restoration in Emotion Recognition[C]//ICASSP 2025-2025 IEEE International Conference on Acoustics, Speech and Signal Processing (ICASSP). IEEE, 2025: 1-5.
> > >
> > > [4] Ding Y, Udompanyawit C, Zhang Y, et al. EEG-based brain-computer interface enables real-time robotic hand control at individual finger level[J]. Nature Communications, 2025, 16(1): 1-20.
> > >
> > > | Model     | SEED    |        |        | SEED-IV |        |        | MI     |        |        |        |
> > > |----------|---------|--------|--------|--------|--------|--------|--------|--------|--------|--------|
> > > |           | 2       | 4      | 8      | 2       | 4      | 8      | 2      | 4      | 8      | 16     |
> > > | ESTformer| 6.9580  | 9.3132 | 9.7285 | 7.1069 | 8.3044 | 8.8623 | 7.0306 |16.6678 |32.3182 |35.7280 |
> > > | STAD     | 9.1939  |11.0372 |14.3967 | 9.5030 |10.9471 |13.1174 | 8.7585 |13.1057 |22.5312 |25.3694 |
> > > | **Ours** | **3.8922** | **5.1186** | **4.9453** | **3.9917** | **4.0779** | **4.8417** | **3.8634** | **7.2958** | **11.5014** | **13.7563** |

---

> > > > ### Author Response · Authors · 2025-11-26
> > > > **Summary of all addressed points**
> > > >
> > > > Thank you very much for your careful review that helped us strengthen both the evaluation protocol and the discussion of generalization.
> > > >
> > > > In the revision, we addressed your main points in the following way
> > > >
> > > > * A more principled justification for using NMSE, PCC and reconstruction SNR in the EEG SR setting.
> > > > * Systematic cross-session and cross-subject experiments are conducted on SEED to better account for variability across sessions and subjects. The results show that, although performance degrades in these harder settings, SRGDiff consistently maintains a clear margin over ESTformer and STAD across all SR factors.
> > > > * We summarized the relative gains of SRGDiff over ESTformer and STAD across datasets and metrics, showing continuous and substantial improvements.
> > > > * A new quantitative metric frequency-domain MAE have been newly introduced to to address the concern about potential selection bias in Figure 4, and SRGDiff achieves the lowest frequency-domain MAE across all datasets and SR scales.
> > > >
> > > > We hope that these additional analyses and clarifications alleviate your concerns about the evaluation protocol and generalization, and we would greatly appreciate it if you could reconsider your assessment of the paper.

---

### Official Review · Reviewer_sP2w · 2025-10-27

**Soundness:** 3
**Presentation:** 3
**Contribution:** 2
**Rating:** 4
**Confidence:** 3

**Summary:**

The work proposes SRGDiff, a diffusion-based approach to solve the task of EEG spatial super-resolution (EEG-SR). SRGDifff proposes two modules, a residual direction module (RDM), predicting per-step residuals from LD EEG features as guidance, and a step-aware modulation module (SMM), providing time-dependent affine scaling and shifting of the latent features.

**Strengths:**

- The work shows thorough experimentation by testing their approach across multiple datasets
- The paper is well-writen and cleanly organized
- The quantitative improvements over previous EEG SR methods are consistent and strong

**Weaknesses:**

Besides their meaningful application to an important domain, the proposed method largely combines existing diffusion-conditioning tricks (residual conditioning, affine modulation, step embeddings). There is little theoretical or conceptual innovation beyond standard conditional diffusion formulations. Overall, the method resembles ControlNet/T2I-Adapter-style modulation, rebranded for EEG. For the ICLR main track, this feels closer to an applied engineering paper than a new machine learning contribution.

**Questions:**

- What is the conceptual difference between SRGDiff’s conditioning mechanism and standard conditional diffusion or ControlNet residual conditioning?
- How does the "residual direction" differ from simply adding predicted corrections in the latent space (i.e., learned noise offsets)?
- The reported FID improvements (Fig. 3) are inconsistent. SRGDiff does not always achieve the lowest FID, and in some settings, baseline methods perform comparably or better. Could the authors detail how the "EEG-FID" is computed, and why it is an appropriate metric for EEG fidelity (which seems primarily reference-based)?
- Did the authors evaluate computational cost against simpler transformer-based SR models (e.g., ESTformer) under equal parameter budgets?

---

> ### Author Response · Authors · 2025-11-20
> **We thank Reviewer sP2w for the effort in reviewing our work and providing supportive comments. We address each concern or question below.**
>
> ### We thank Reviewer sP2w for the effort in reviewing our work and providing supportive comments. We address each concern or question below.
>
> > W1: Besides their meaningful application to an important domain, the proposed method largely combines existing diffusion-conditioning tricks (residual conditioning, affine modulation, step embeddings). There is little theoretical or conceptual innovation beyond standard conditional diffusion formulations. Overall, the method resembles ControlNet/T2I-Adapter-style modulation, rebranded for EEG.
> >
> > Q1: What is the conceptual difference between SRGDiff’s conditioning mechanism and standard conditional diffusion or ControlNet residual conditioning?
>
> **Response.** We thank the reviewer for the constructive comments and for recognizing the relevance of the application domain. At a high level, controlNet learns generic feature maps without explicit link to conditional posterior, whereas SRGDiff explicitly approximate $\mu (z_\theta | X^L)$ stepwise and plug it into the DDPM posterior formula. To our knowledge, most residual conditioning schemes, e.g., ControlNet-style, treat the residuals as additional feature maps trained via the standard noise-prediction loss [1-3]. These residual features are not relevant to how the reverse diffusion trajectory should evolve conditioned on the input. In contrast, EEG SR involves a different relationship between the low-density input and the desired high-density reconstruction. The low-density signal is a partial observation of the same underlying high-density process, where channels are related by the physical sensor montage. Our conditioning mechanism is therefore designed to predict the HD latent residual from the LD observation and injects this residual as an explicit correction to the reverse update.
>
> Conceptually, the key difference from ControlNet-style conditioning is that our residual head is explicitly supervised to approximate the time-scaled conditional mean increment implied by the partially observed low-density EEG. ControlNet injects generic condition features into the U-Net and lets the model implicitly discover how to use them, while our residual diffusion module is trained to follow the Bayes-optimal reconstruction direction under the assumed linear Gaussian generative model. For completeness, we outline the derivation below.
>
> We first formalize the LD–HD relation and the latentization:
> $$
> X^L = M X^H + \xi,\quad \xi\sim\mathcal N(0,\Sigma_\xi). \tag{1}
> $$
> $$
> z_0 = E(X^H),\qquad X^H \approx D(z_0). \tag{2}
> $$
> where $M$ is a fixed channel-selection operator and $\xi$ is noise, and we are interested in the conditional distribution $p(X^{H} \mid X^{L})$.
>
> Given $z_0$, the forward diffusion is
> $$
> z_t = \alpha_t z_0 + \sigma_t \epsilon,\quad \epsilon\sim\mathcal N(0,I),\ t=1,\dots,T. \tag{3}
> $$
>
> This induces the diffusion-path residual
> $$
> \delta z_t := z_0 - z_t = (1-\alpha_t)z_0 - \sigma_t \epsilon. \tag{4}
> $$
>
> Our residual head $ R_\varphi $ is trained with a squared loss,
> $$
> L_{\mathrm{res}}=\mathbb{E} \left[ ||R_\varphi(X^L,t)-\delta z_t ||_2^2 \right]. \tag{5}
> $$
>
> which, by the optimality of squared loss, targets the conditional mean
> $$
> R_\varphi^*(X^L,t)=\mathbb{E}[\delta z_t\mid X^L,t]. \tag{6}
> $$
>
> Combining (4) and $\mathbb{E}[\epsilon\mid X^L,t]=0$ gives the analytic form of this target:
> $$
> \mathbb{E}[\delta z_t\mid X^L,t]=(1-\alpha_t) \mathbb{E}[z_0\mid X^L]. \tag{7}
> $$
>
> Thus the residual head learns a time-scaled conditional posterior mean of the HD latent, rather than an unconstrained feature.
>
> We now connect this supervision to the reverse kernels. For DDPM, the exact one-step posterior mean is
> $$
> \mu^*(z_t,z_0)=a_t z_t + b_t z_0. \tag{8}
> $$
>
> Conditioning on $X^L$ yields
> $$
> \mu^*(z_t,X^L)=a_t z_t + b_t \mathbb{E}[z_0\mid X^L]. \tag{9}
> $$
> We parameterize the learned reverse mean as
> $$
> \mu_\theta(z_t,X^L,t)=a_t z_t + c_t R_\varphi(X^L,t). \tag{10}
> $$
>
> Combining (7) with the choice
> $$
> c_t=\frac{b_t}{1-\alpha_t,} \tag{11}
> $$
> gives $\mu_\theta\approx\mu^*$. Hence, with an appropriate variance, the learned kernel satisfies
> $$
> p_\theta(z_{t-1}\mid z_t,X^L)\ \approx\ p(z_{t-1}\mid z_t,X^L)\quad\text{for all }t. \tag{12}
> $$
>
> By induction from the true $p(z_T\mid X^L)$, the multi-step reverse process yields
> $$
> z_0 \sim p_\theta(z_0\mid X^L)\ \approx\ p(z_0\mid X^L). \tag{13}
> $$
>
> Finally, decoding realizes conditional HD reconstruction: sampling with (10)–(11) yields (z_0^{(m)}\sim p_\theta(z_0\mid X^L)) and
> $$
> X^{H(m)}=D\big(z_0^{(m)}\big)\ \sim\ p(X^H\mid X^L),\qquad
> \widehat{\mathbb{E}}[X^H\mid X^L]=\tfrac1M\sum_m D(z_0^{(m)}). \tag{14}
> $$
>
> To Continue

---

> > ### Author Response · Authors · 2025-11-20
> > **Continued response (2/3)**
> >
> > > W1: Besides their meaningful application to an important domain, the proposed method largely combines existing diffusion-conditioning tricks (residual conditioning, affine modulation, step embeddings). There is little theoretical or conceptual innovation beyond standard conditional diffusion formulations. Overall, the method resembles ControlNet/T2I-Adapter-style modulation, rebranded for EEG.
> > >
> > > Q1: What is the conceptual difference between SRGDiff’s conditioning mechanism and standard conditional diffusion or ControlNet residual conditioning?
> >
> > Reference
> >
> > [1] Zhang L, Rao A, Agrawala M. Adding conditional control to text-to-image diffusion models[C]//Proceedings of the IEEE/CVF international conference on computer vision. 2023: 3836-3847.
> >
> > [2] Mou C, Wang X, Xie L, et al. T2i-adapter: Learning adapters to dig out more controllable ability for text-to-image diffusion models[C]//Proceedings of the AAAI conference on artificial intelligence. 2024, 38(5): 4296-4304.
> >
> > [3] Zhao S, Chen D, Chen Y C, et al. Uni-controlnet: All-in-one control to text-to-image diffusion models[J]. Advances in Neural Information Processing Systems, 2023, 36: 11127-11150.
> >
> > > Q2: How does the "residual direction" differ from simply adding predicted corrections in the latent space (i.e., learned noise offsets)?
> >
> > **Response.** We thank the reviewer for the question. Our residual direction is an explicitly supervised, timestep-dependent reconstruction direction derived from the forward diffusion process under partial observation, rather than an unconstrained latent noise offset.
> >
> > (1) **Definition and supervision.** We explicitly define the forward residual at each timestep $t$ as
> > $$
> > \delta z_t = z_0 - z_t,
> > $$
> > where $z_t$ follows the known forward diffusion and $z_0 = E(X^{H})$. The guidance network is trained to approximate this analytic residual, conditioned on $(z_t, h^L, t)$, so the learned direction is directly supervised to align with the forward noising trajectory under the partial observation $X^{L}$.
> >
> > (2) **Time-varying trajectory alignment.** The condition defines a time-varying, trajectory-aligned vector field over the diffusion path, rather than a single static correction reused across steps. This step-wise refinement is particularly important in the partial-observation setting $X^{L} = M X^{H} + \xi$, where the missing HD information must be progressively recovered along the trajectory instead of being injected as a one-shot latent offset.
> >
> >
> > > Q3: The reported FID improvements (Fig. 3) are inconsistent. SRGDiff does not always achieve the lowest FID, and in some settings, baseline methods perform comparably or better. Could the authors detail how the "EEG-FID" is computed, and why it is an appropriate metric for EEG fidelity (which seems primarily reference-based)?
> >
> > **Response.** After carefully re-checking both Fig. 3 and the underlying numbers, we confirm that SRGDiff (light blue square) achieves the lowest EEG-FID across all reported settings; there is no case where a baseline outperforms SRGDiff in EEG-FID in this figure. If anything remains unclear, we would be happy to provide further details.
> >
> > Regarding the definition of EEG-FID, we adopt an EEG-FID analogous to image FID, as in prior time-series work DiffuSETS (ECG). Concretely, we train and freeze an EEG-specific encoder (EEGNet) on the training split of each dataset. Then, we extract embeddings on the test split using the pretrained EEGNet for (i) real HD EEG and (ii) reconstructed EEG. Finally, we fit two multivariate Gaussians on these embeddings and compute the Fréchet distance between them. A lower EEG-FID therefore means that reconstructed signals follow the same feature distribution as real HD signals on that dataset.
> >
> > We acknowledge that EEG-FID is less theoretically grounded than image FID [1], since it relies on a learned EEG encoder and its value can depend on this choice. However, using FID-style distances in a task-aware feature space has become a common practice for evaluating generative time-series models [2-3] (e.g., ECG and other time series signals), and we adopt EEG-FID in the same spirit: as a complementary, distribution-level measure of fidelity alongside the reference-based signal metrics (NMSE/PCC/SNR) that we already report.
> >
> > Reference
> >
> > [1] Heusel M, Ramsauer H, Unterthiner T, et al. Gans trained by a two time-scale update rule converge to a local nash equilibrium[J]. Advances in neural information processing systems, 2017, 30.
> >
> > [2] Lai Y, Chen J, Zhao Q, et al. DiffuSETS: 12-Lead ECG generation conditioned on clinical text reports and patient-specific information[J]. Patterns, 2025.
> >
> > [3] Kong Z, Ping W, Huang J, et al. DiffWave: A Versatile Diffusion Model for Audio Synthesis[C]//International Conference on Learning Representations.

---

> > > ### Author Response · Authors · 2025-11-20
> > > **Continued response (3/3)**
> > >
> > > > Q4: Did the authors evaluate computational cost against simpler transformer-based SR models (e.g., ESTformer) under equal parameter budgets?
> > >
> > > **Response.** Thank you for raising this point. In the original draft we had mainly reported runtime for each method (Table 10), showing that SRGDiff already satisfies real-time constraints for EEG SR. Following your suggestion, we have added a more detailed comparison of computational cost under comparable parameter budgets, summarized below. For ESTformer, STAD, and SRGDiff we report both the total number of trainable parameters and the theoretical computation cost per input window, measured as the number of floating-point operations (GFLOPs) for a fixed 4s window length.
> > >
> > > As shown in the table, compared with ESTformer and STAD, SRGDiff has a markedly smaller parameter count and lower per-window FLOPs. ESTformer relies heavily on multi-head self-attention layers, and STAD uses sizeable MLP blocks, both of which contribute substantial parameter counts and quadratic or dense computation. In contrast, SRGDiff is built on a predominantly convolutional backbone with lightweight conditioning modules (RDM/SMM), which keeps both the parameter count and per-window GFLOPs significantly lower. We include this table and a short discussion in the revised manuscript under Subsection G.1 to clarify the computational trade-offs between SRGDiff and simpler transformer-based SR models.
> > >
> > >
> > > | Model    | \#Params (M) | FLOPs / window (G) |
> > > |--------- |-------------:|-------------------:|
> > > | ESTformer | 12.111      | 4.302              |
> > > | STAD      | 13.949      | 1.650              |
> > > | SRGDiff (ours)  | 2.342       | 1.380              |

---

> > > > ### Author Response · Authors · 2025-11-26
> > > > **Summary of all addressed points**
> > > >
> > > > Thank you again for your valuable feedback that helped us to significantly improve our paper.
> > > >
> > > > More concretely, we believe we have addressed your main concerns as follows
> > > >
> > > > * A detailed conceptual and mathematical discussion comparing our conditioning mechanism with standard conditional diffusion and ControlNet-style residual conditioning.
> > > > * Clarification about how the proposed residual direction differs from simply adding predicted latent noise offsets.
> > > > * We re-checked the EEG-FID results and confirmed that SRGDiff achieves the best EEG-FID across all reported settings. We also added a precise definition of EEG-FID and discussed why it serves as a complementary distribution-level metric alongside reference-based signal metrics.
> > > > * Including a new comparison of computational costs by reporting both parameter counts and FLOPs per 4-second window for ESTformer, STAD, and SRGDiff.
> > > >
> > > > We hope these additions clarify the conceptual distinctions and practical trade-offs, and we would be very grateful if you could kindly reassess your score in view of the revised manuscript.

---

### Official Review · Reviewer_7K5e · 2025-10-31

**Soundness:** 2
**Presentation:** 3
**Contribution:** 3
**Rating:** 6
**Confidence:** 4

**Summary:**

This paper presents SRGDiff, a novel diffusion-based method for spatial super-resolution of EEG that addresses the consistency–fidelity trade-off by dynamically conditioning the reverse process on low-density EEG data using residual guidance. Specifically, the authors propose using the forward-noising residual from low-density channels as a per-step corrective direction in the denoising. To achieve this, a Residual Direction Module (RDM) predicts a path residual for directional correction, and a Step-Aware Modulation Module (SMM) predicts scale and bias for calibration of the residual update.

Extensive experiments on three datasets (SEED, SEED-IV, and Localize-MI) demonstrate that SRGDiff outperforms strong baselines (e.g., ESTformer, STAD, DDPMEEG) across signal-, feature-, and downstream-level evaluations over multiple upsampling scales.

**Strengths:**

Originality. The idea of dynamically guiding each denoising step with a residual derived from the LD forward process is novel and conceptually distinct from prior approaches that rely on static conditioning.

Quality. The method is comprehensively evaluated across three public datasets and multiple SR scales, covering signal-, feature-, and downstream-level tasks. Ablations demonstrate the importance of both RDM and SMM.

Significance. SRGDiff consistently achieves superior performance measured by NMSE, PCC, and SNR compared to strong baselines. The improvements are meaningful and practically relevant for EEG-based BCI applications.

Clarity. The paper is well-structured, concise, and coherent. Figures effectively illustrate the model components and results.

**Weaknesses:**

- The authors claim that SRGDiff is transferable to general SR settings, while all experiments are restricted to EEG. This should at least be discussed in more detail but ideally be experimentally verified to support the claim.
- The impact of $\lambda_{res}$ and $\lambda_{SMM}$ is not studied. Adding an experiment to the ablation section where $\lambda_{res}$ and $\lambda_{SMM}$ are varied would provide insight into the sensitivity and impact of hyperparameters.
- In the abstract, the claim of “up to 40% gains” over baselines is vague. It would improve clarity to specify under which dataset, SR scale, and metric these gains occur.
- When the authors or the publication are not included in the sentence, the citation should be in parenthesis using \citep{}, as outlined in the formatting instructions.
- Some abbreviations are not defined (e.g., BCI, PCC).
  Figure 4 is not readable due to the text being too small.

If the authors address the weaknesses the reviewer would be willing to increase the rating.

**Questions:**

- How was the form of the RDM and SMM chosen? Were other forms explored?
- Is SRGDiff applicable to variable or irregular LD electrode layouts? For example, could a model trained on 16 electrodes generalize to 8-electrode inputs, or would retraining be required?
- Have the authors experimented with SR tasks beyond EEG to support the claim of generality?

---

> ### Author Response · Authors · 2025-11-20
> **We thank Reviewer 7K5e for the effort in reviewing our work and providing supportive comments. We address each concern or question below.**
>
> > W1: The authors claim that SRGDiff is transferable to general SR settings, while all experiments are restricted to EEG.
> >
> > Q3: Have the authors experimented with SR tasks beyond EEG to support the claim of generality?
>
> **Response.** Thank you for raising this point. We agree that our original wording *“transferable to general SR settings”* was too broad. Our intention was not to claim applicability to arbitrary super-resolution problems, but to different EEG datasets and a range of SR factors (e.g., 2×/4×/8×) within our domain. In the revised manuscript, we therefore replace “general SR settings” with "remains effective across EEG datasets and a wide range of SR factors.".
>
> To directly address your concern about application potential, we further evaluate SRGDiff on invasive ECoG recordings from subject 7 of the AJILE12 dataset [1], following the experimental setup of [2] (same dataset, windowing protocol, and train/validation/test split).
> We compare against ESTformer and STAD under $2\times$, $4\times$, $8\times$ settings. The reconstruction error is summarized below and the full details are included in the revised manuscript under Subsection H.2.
>
> References
>
> [1] Peterson S M, Singh S H, Dichter B, et al. AJILE12: Long-term naturalistic human intracranial neural recordings and pose[J]. Scientific data, 2022, 9(1): 184.
>
> [2] Vetter J, Macke J H, Gao R. Generating realistic neurophysiological time series with denoising diffusion probabilistic models[J]. Patterns, 2024, 5(9).
>
> | Model   | Metric | 2×     | 4×     | 8×     |
> |---------|--------|:------:|:------:|:------:|
> | ESTformer | NMSE | 0.4573 | 0.7189 | 0.8517 |
> |         | PCC   | 0.7367 | 0.5299 | 0.3845 |
> |         | SNR   | 3.3991 | 1.4334 | 0.6974 |
> | STAD    | NMSE  | 0.4932 | 0.6901 | 0.7987 |
> |         | PCC   | 0.6854 | 0.5118 | 0.4312 |
> |         | SNR   | 3.1686 | 1.4449 | 1.1684 |
> | SRGDiff | NMSE  | 0.3575 | 0.6529 | 0.7312 |
> |         | PCC   | 0.8023 | 0.5332 | 0.4502 |
> |         | SNR   | 4.8913 | 2.1657 | 1.9089 |
>
>
> > W2: The impact of λ_res and λ_SMM is not studied. Adding an experiment to the ablation section where λ_res and λ_SMM are varied would provide insight into the sensitivity and impact of hyperparameters.
>
> **Response.** Thank you for the suggestion. We have added an ablation study varying $\lambda_{\text{res}}$ and $\lambda_{\text{SMM}}$ on all three datasets under the most challenging SR setting. In our training objective, $\lambda_{\text{res}}$ scales the residual-direction supervision term for the residual diffusion module (RDM), and $\lambda_{\text{SMM}}$ scales the step-aware modulation regularizer for SMM. Their main purpose is to put the three loss components (standard diffusion reconstruction loss, residual supervision loss, and SMM regularization) on a comparable numerical scale, so that none of them dominates the optimization purely due to magnitude.
>
> To verify this, we report below the results for each dataset in the settings of highest SR factor. Concretely, we sweep $ \lambda_{\text{res}} \in {0.1, 0.5, 1.0, 2.0, 5.0} $ (relative to the default), and $ \lambda_{\text{SMM}} \in {0.001, 0.005, 0.01, 0.02, 0.1} $. All numbers in the tables correspond to NMSE (lower is better). The results shows that SRGDiff remains relatively robust within a broad range of values. Further details are included in the revised manuscript in Subsection D.3.
>
> | $\lambda_{\text{res}}$ | SEED   | SEED IV | Localize-MI |
> | ---------------------- | ------ | ------- | ----------- |
> | 0.1   | 0.3928 | 0.3286   | 0.3941      |
> | 0.5   | 0.3532 | 0.2822   | 0.3578      |
> | 1.0   | 0.3494 | 0.2603 | 0.3457   |
> | 2.0   | 0.3508 | 0.2810   | 0.3565      |
> | 5.0   | 0.4012 | 0.3369   | 0.3827      |
>
> | $\lambda_{\text{SMM}}$ | SEED   | SEED IV | Localize-MI |
> | ---------------------- | ------ | ------- | ----------- |
> | 0.001  | 0.3975 | 0.3133   | 0.3904      |
> | 0.005  | 0.3539 | 0.2696   | 0.3519      |
> | 0.01 | 0.3494 | 0.2603 | 0.3457 |
> | 0.02   | 0.3513 | 0.2712   | 0.3489      |
> | 0.1    | 0.3990 | 0.3240   | 0.3915      |

---

> > ### Author Response · Authors · 2025-11-20
> > **Continued response (2/3)**
> >
> > > W3: In the abstract, the claim of “up to 40% gains” over baselines is vague. It would improve clarity to specify under which dataset, SR scale, and metric these gains occur.
> >
> > **Response.** We appreciate this remark and have revised the abstract to be more precise. We now write:
> >
> > *SRGDiff consistently achieves higher SNR than the baseline ESTformer and STAD among Localize-MI, SEED and SEED-IV datasets, with up to roughly $75\%$ relative SNR improvement in the most challenging $8\times$ setting.*
> >
> > > W4: When the authors or the publication are not included in the sentence, the citation should be in parenthesis using \citep{}, as outlined in the formatting instructions.
> >
> > **Response.** Thank you for the formatting reminder. We have revised all relevant instances to parenthetical citations using `\citep{}` at:
> >
> > * Introduction (L38–43, L51–53, L60–72),
> > * Related Works (L107–131, L136–142),
> > * Experiment (L279–286, L308–313),
> > * Appendix (L635, L689).
> >
> >
> > > W5: Some abbreviations are not defined (e.g., BCI, PCC). Figure 4 is not readable due to the text being too small.
> >
> > **Response.** Thank you for pointing this out. We have now explicitly defined all relevant abbreviations at their first occurrence, including:
> >
> > * STFT (short-time Fourier transform),
> > * PCC (Pearson correlation coefficient),
> > * BCI (brain–computer interface),
> > * NMSE (normalized mean squared error),
> > * SNR (signal-to-noise ratio with respect to the HD reference).
> >
> > In addition, we have revised Figure 4 by changing the layout from three to two topomaps per row, with larger fonts and higher resolution so that all labels and topographies are clearly readable.
> >
> >
> > > Q1: How was the form of the RDM and SMM chosen? Were other forms explored?
> >
> > **Response.** Thank you for the question. In the paper we adopt a specific residual and modulation form for RDM and SMM, and we have also explored several alternative architectures; below we first restate the chosen forms for clarity and then summarize the ablation results.
> >
> > The concrete forms of RDM and SMM are explicitly defined in Sec. 4.2 and Sec. 4.3 of the main paper; for clarity, we briefly restate their structures here.
> >
> > **RDM (Residual Direction Module)** is implemented as a lightweight convolutional / MLP predictor that takes the LD features \(c\) and timestep embedding \(\tau(t)\) as input and outputs a residual feature $R_{\phi}(c, t)$. This residual is supervised using the analytically defined forward-noising residual $\delta z_t = z_0 - z_t$ and is additively fused into the latent state:
> >
> > $$
> > \hat{z}^{\text{RDM}}_t = \text{LayerNorm}(\hat{z}_t) + R\_{\phi}(c, t).
> > $$
> >
> > **SMM (Step-aware Modulation Module)** takes the same conditioning and produces channel-wise affine parameters $ (\gamma_t, \beta_t) $ via two MLP heads, which modulate the RDM-updated state in a Feature-wise Linear Modulation-style manner:
> >
> > $$
> > \hat{z}^{\text{SMM}}_t = \gamma_t \odot \hat{z}^{\text{RDM}}_t + \beta_t.
> > $$
> >
> > We chose this residual + affine modulation design because it (i) introduces minimal architectural overhead to the backbone U-Net, (ii) provides stable per-step conditioning, and (iii) directly aligns with the analytical residuals from the forward noising process.
> >
> > To validate these design choices, we conducted ablations that target two aspects:
> > (i) how the residual direction from LD is injected into the latent trajectory, and
> > (ii) how step-aware modulation is parameterized.
> >
> > * A1 (Conv1d fusion for RDM) and A2 (cross-attention fusion for RDM) keep SMM unchanged but replace our simple additive RDM fusion. A1 concatenates the residual feature with the current latent state $ z_t $ and fuses them via a 1D convolution; A2 treats the residual as key/value and $ z_t $ as query in a lightweight cross-attention block. These variants test whether more complex fusion mechanisms can outperform the proposed additive residual direction.
> >
> > * A3 (no gate for SMM) and A4 (static gate for SMM) keep RDM unchanged but modify SMM. A3 directly adds LD features to the diffusion features without any learned modulation. A4 uses a timestep-independent affine gate $ (\gamma, \beta) $ shared across diffusion steps. These variants test whether our step-aware FiLM-style modulation is really necessary compared with no modulation or a static one.
> >
> > All values in the table below report NMSE (lower is better). We observe that SRGDiff consistently achieves lower NMSE than all these alternatives, suggesting that additive residual direction and step-dependent FiLM modulation are both important for the final performance.
> >
> > | Model            | SEED   | SEED-IV | Localize-MI |
> > |------------------|--------|---------|-------------|
> > | A1 (concat)      | 0.3725 | 0.3651  | 0.3688      |
> > | A2 (cross-attn)  | 0.3613 | 0.3194  | 0.3581      |
> > | A3 (no gate)     | 0.4009 | 0.3735  | 0.4362      |
> > | A4 (static gate) | 0.3742 | 0.3360  | 0.3809      |
> > | SRGDiff          | 0.3494 | 0.2603  | 0.3457      |

---

> > > ### Author Response · Authors · 2025-11-20
> > > **Continued response (3/3)**
> > >
> > > > Q2: Is SRGDiff applicable to variable or irregular LD electrode layouts? For example, could a model trained on 16 electrodes generalize to 8-electrode inputs, or would retraining be required?
> > >
> > > **Response.** We appreciate this practical question for real-world deployment. In fact, SRGDiff can handle irregular LD layouts at inference time without retraining. During inference, the low-density input are first mapped into a common spatial representation via pretrained VAE, so it is not tightly coupled to a specific LD montage.
> > >
> > > To verify this empirically, we trained SRGDiff with a 16-electrode LD configuration in SEED and evaluated it at test time on 8/10/12/14-electrode inputs obtained by subsampling the LD montage. In a separate experiment, we also trained SRGDiff with a 32-electrode LD configuration and evaluated it on 8- and 16-electrode inputs, again using only subsampling at test time and no retraining. Without any retraining, the model maintains good reconstruction quality, although there is an expected performance drop under more severe undersampling. More details are included in the revised manuscript in Subsection H.3.
> > >
> > > | Train LD channels | Test LD channels | NMSE↓   | PCC↑   | SNR↑   |
> > > |-------------------|------------------|--------|--------|--------|
> > > | 16                | 8                | 0.4542 | 0.7196 | 4.0329  |
> > > | 16                | 10               | 0.4031 | 0.7650 | 4.4211  |
> > > | 16                | 12               | 0.3588 | 0.8012 | 4.8705  |
> > > | 16                | 14               | 0.3245 | 0.8268 | 5.1203  |
> > > | **16**            | **16 (base)**   | **0.2977** | **0.8445** | **5.2606** |
> > > | 32                | 8                | 0.4753 | 0.6735 | 3.9518  |
> > > | 32                | 16               | 0.3585 | 0.7820 | 4.4002  |
> > > | **32**            | **32 (base)**   | **0.1632** | **0.9102** | **7.8413** |

---

> > > > ### Author Response · Authors · 2025-11-26
> > > > **Summary of all addressed points**
> > > >
> > > > Thank you again for your thoughtful review, which helped us substantially improve the paper.
> > > >
> > > > Specifically, we believe we have addressed your main points in the following way
> > > >
> > > > * The original claim of “transferable to general SR settings” is reclaimed and SRGDiff have been additionally evaluated on invasive ECoG recordings (AJILE12) and compared it against ESTformer and STAD.
> > > > * The robustness of hyperparameters $ λ_ {res}$ and $λ_ {SMM}$ have been demonstrated with an additional ablation study, showing that SRGDiff remains robust over a broad range of values without per-dataset fine-tuning.
> > > > * The proposed modules RDM and SMM are evaluated with ablations of alternative residual fusion and modulation forms, demonstrating the chosen additive residual + step-aware FiLM design.
> > > > * We added experiments on variable and irregular LD electrode layouts, where a model trained on one LD configuration is evaluated on subsampled layouts without retraining, demonstrating robustness to missing channels.
> > > >
> > > > In the light of these substantial revisions, we would be very grateful if you could kindly reconsider your overall evaluation of the paper.

---

### Author Response · Authors · 2025-11-20
**General response**

We sincerely thank all reviewers for their thoughtful comments and constructive feedback, which have significantly improved our work. In the revised manuscript, we have comprehensively addressed each concern, incorporating additional experiments, theoretical analyses, and detailed discussions. Major changes and updates are highlighted in blue for your convenience.

Please do not hesitate to let us know if there are any additional concerns or suggestions. We are happy to provide further clarifications or make further revisions if needed.

Thank you once again for your valuable time and insights.

---

### Author Response · Authors · 2025-12-01
**Summary of reviewer status and request for area chair consideration**

Dear area chairs,

Thank you for your time in handling our submission.

Below we summarize how we addressed the main concerns raised by the reviewers and the current review status.


# Response to reviewer 7K5e

In our latest responses to reviewer 7K5e, we addressed their main concerns as follows:

* Evaluation of SRGDiff on the new invasive ECoG dataset following the setup of Vetter et al., showing consistent gains over ESTformer and STAD.
* Evaluation of ablation model on two hyper-parameters λ_res and λ_SMM, showing that SRGDiff is robust across a broad range and explaining how extreme values reduce to degenerate variants of the model.
* Evaluation of the proposed modules RDM and SMM, adding ablations comparing against more complex fusion (Conv1d / cross-attention) and different modulation schemes (no gate / static gate), confirming that our simple additive residual plus step-dependent FiLM modulation provides the best trade-off.
* Paper clarity and presentation polishment.

**Reviewer 7K5e (rating:6, confidence:4) explicitly stated that “If the authors address the weaknesses, the reviewer would be willing to increase the rating.”** Since we believe we have addressed the main points raised by reviewer 7K5e in our revision, we respectfully ask the area chair to consider a reassessment of the rating of Reviewer 7K5e on our paper.


# Response to reviewer sP2w

For reviewer sP2w, we substantially strengthened the conceptual and theoretical exposition and clarified the metrics and efficiency:

* Clarification on conceptual difference from ControlNet / T2I-Adapter.
* Discussion of residual direction vs. latent noise offsets.
* Clarification EEG-FID definition and behavior.
* Aditional experiment on computational cost vs. transformers, showing that SRGDiff uses substantially fewer parameters and lower FLOPs while meeting real-time constraints.

**According to the official ICLR Pangram AI-content dashboard for submission 3877, this is the only review (rating:4, confidence:3) marked as “Fully AI-generated”, while the other three are only “Lightly AI-edited”; this public dashboard is fully anonymized and does not violate double-blind review: [https://iclr.pangram.com/reviews?submission_number=3877](https://iclr.pangram.com/reviews?submission_number=3877)**


# Response to reviewer cWrT

In our latest responses to reviewer cWrT, we addressed their main concerns as follows:

* Justification of NMSE, PCC, and SNR. We clarified that our SNR is a reconstruction SNR w.r.t. the HD reference, so better reconstructions necessarily yield higher SNR.
* Additional cross-subject and cross-session evaluation, showing that SRGDiff maintains a consistent margin over ESTformer and STAD under both types of shifts.
* Discussion of our main contribution.

**Reviewer cWrT (confidence:4) increased their score from 4 to 6 on Nov 26 around 23:00, before the information-leak issue occurred;** while he/she have not yet posted a follow-up textual comment after this score update.



# Response to reviewer pkZv

In our responses to reviewer pkZv, we focused on clarifying the role of static guidance and the ablation baselines:

* Clarification on the “LDM” baseline.
* Discussion of our contributions.
* Justification on the gains between ablation models and other baselines.
* Additional evaluation on the runtime of our ablation models.

Reviewer pkZv maintains a positive overall score of 6 with confidence 4.

We hope that the area chair considers this context in their evaluation of this 3877 paper.

Thank you!

---

### Meta-Review · Area_Chair_caok · 2026-01-06

**Summary:**

This paper aims to use a conditionally diffusion model to achieve EEG spatial super-resolution from a cost-effective, sparse sensor array. The model, dubbed SRGDiff, employs a dynamic conditional generation strategy rather than the conventional static conditioning features. This could be a unique contribution. Moreover, SRGDiff utilizes a Residual Direction Module (RDM) to predict stepwise differences and a Step-Aware Modulation Module (SMM) for time-dependent affine transformations. Experiments were conducted on SEED, SEED-IV, and Localize-MI datasets. During the rebuttal, the authors extended the evaluation to invasive ECoG data (AJILE12) to demonstrate generalization.

**Reviewer Concerns:**

1. Reviewers SP2w raise significant concerns over the conceptual novelty of SRGDiff compared with prior diffusion conditioning. In rebuttals, the author explained that SRGDiff’s residual direction is unique from other designs. They used mathematical and conceptual explanations to demonstrate that the residual direction is analogous to the timestep-dependent conditional posterior mean. Therefore, such designs distinguish SRGDiff from generic residual feature injection of ControlNet.
2. reviewer pkZv pointed out there are issues in ablation studies. In response, the authors clarified that the “LDM” baseline already uses static LD guidance. To avoid confusion, they renamed it to “LDM+LD” and further added explicit comparisons. Additioanl runtime and FLOPs analyses are also provided.
3.  reviewer cWrT asked about the metric justification.  The responses justified NMSE, PCC, and reconstruction SNR using a conditional estimation viewpoint. They also introduced frequency-domain MAE.

 Theoretical guarantees remain limited, but this was not a strict expectation for applied generative works like this paper.

**Reviewer Scores:**

Reviewer 7K5e: would keep above 6 since the concerns on generalization were addressed with new experiments.

Reviewer cWrT: would raise to 6 from 4 since the validation concerns were addressed.

Reviewer pkZv: keep positive 6

Reviewer sP2w: would raise to 6 from 4 since the technical concerns on novelty and "ControlNet similarity" were addressed

---

### Decision · Program_Chairs · 2026-01-26

Accept (Poster)